# tRNA-derived RNA processing in sperm transmits non-genetically inherited phenotypes to offspring in *C. elegans*

Nicholas S. Galambos[1,2,3,5], Olivia J. Crocker[1,2,5], Blair K. Schneider[1,2], Kennelly S. Allerton[2], Katrin E. Gross[1,2], Alexandra E. Schneider[1,2], Jared Lynch [2], Onur Yukselen[4], Alper Kucukural [4] & Colin C. Conine [1,2] ✉

The environment encountered by an organism can modulate epigenetic information in gametes to transmit non-genetically inherited phenotypes to offspring. In mouse models, the diet of males regulates specific tRNA-derived RNAs (tDRs) in sperm. After fertilization, tDRs regulate embryonic gene expression and generate metabolic phenotypes in adult offspring through uncharacterized changes during development. Here we demonstrate that in *C. elegans*, tDRs also accumulate in sperm and similarly transmit epigenetically inherited phenotypes to offspring. We identify the RNaseT2 enzyme, *rnst-2*, as a regulator of *C. elegans* tDR accumulation. RNST-2 processes or degrades tRNA-halves, to short <30 nt fragments. This *rnst-2* dependent regulation of tDR length distribution modulates specific tDRs in sperm which, after fertilization, regulate early embryonic and developmental gene expression, leading to adaptive phenotypes in progeny. Our findings establish tDRs as a conserved carrier of intergenerational epigenetic information and position the worm as a model for dissecting paternal non-genetic inheritance mechanistically.

To successfully develop and reproduce, organisms must adapt to continuously changing environmental conditions, which include, but are not limited to, nutrient availability, temperature, predation, and mating/courtship. The ability to successfully adapt to changing environments was long thought to predominantly occur through natural selection of the fittest phenotype manifested from the genotype, or DNA of the organism, over many generations. However, it is now widely recognized that phenotypes are influenced not only by genetic sequence but also by the inherited epigenetic regulation of the genome. Indeed, it has now been demonstrated in numerous model organisms ranging from *C. elegans* to rodents that the ancestral environment of both mothers and fathers can regulate epigenetic information in the gametes and thereby transmit non-genetically inherited

phenotypes to progeny[1–3]. While canonical modes of epigenetic regulation, such as DNA methylation and histone post-translational modifications, can be modulated by the environment and have the potential to transmit inherited phenotypes, definitive experimental evidence for inheritance through these molecules has been sparse. Conversely, small regulatory RNAs in sperm, including both miRNAs and tRNA-derived RNAs (tDRs), have been demonstrated to causally communicate inherited phenotypes from father to offspring through zygotic RNA microinjection studies in mammals[4–7]. Additionally, in *C. elegans*, endogenous small RNA and RNA interference (RNAi) pathways in sperm and oocytes have been shown to transmit non-genetically inherited phenotypes both intergenerationally to F1 progeny, as well as transgenerationally to the F2 generation and beyond[8–12].

[1]Division of Neonatology, Children's Hospital of Philadelphia, Philadelphia, PA, USA. [2]Departments of Genetics and Pediatrics - Penn Epigenetics Institute, Institute of Regenerative Medicine, and Center for Women's Health and Reproduction Medicine, University of Pennsylvania Perelman School of Medicine, Philadelphia, PA, USA. [3]Department of Biology, University of Pennsylvania, School of Arts & Sciences, Philadelphia, PA, USA. [4]Department of Genomics and Computational Biology, University of Massachusetts Chan Medical School, Worcester, MA, USA. [5]These authors contributed equally: Nicholas S. Galambos, Olivia J. Crocker. ✉e-mail: conine@upenn.edu

Eukaryotic small regulatory RNAs, or simply small RNAs, are 18–40 nucleotide (nt) RNAs that predominantly function in RNAi pathways. These include endogenous pathways such as the piwi-interacting RNA (piRNA), microRNA (miRNA), and endogenous small interfering RNA (endo-siRNA) pathways, as well as the exogenous RNAi pathway wherein exposure of an organism to double-stranded RNA (dsRNA) triggers sequence-specific degradation of endogenous RNAs with complementary sequence to the dsRNA. RNAi and miRNAs were discovered and genetically characterized in *C. elegans*, with small RNA function first being described in plants and further biochemically characterized in *Drosophila*[13–17]. More recently, a new class of eukaryotic small RNAs with gene regulatory functions has been identified. Often referred to as tRNA-fragments (tRFs) or tDRs, these small RNAs accumulate as a result of the endonucleolytic cleavage or degradation of tRNAs into smaller RNA species. tDRs can accumulate as tRNA-halves, which are 5′ or 3′ pieces of tRNAs cut at the anticodon loop, ranging in size from 30 to 40 nt. Additionally, small <30 nt fragments have been identified from the 5′, 3′, or interior portion of the mature tRNA sequence. While tDRs have been demonstrated to have a plethora of cellular functions[18–23], including operating as piRNAs in ciliates[24,25], and interfering with retroviral transcription in murine models[5,26,27], their functions have mostly been described in the context of stress in mammalian cell culture and yeast. In both model systems, oxidative stress, stress responses from exposure to arsenic, heat/cold shock, and UV radiation result in production of tRNA-halves via cleavage of full-length tRNAs at the anticodon loop[28,29]. The endonucleolytic cleavage required to make tRNA-halves is dependent on RNaseA enzymes, including Angiogenin, in mammalian cells and the RNaseT2 enzyme Rny1 in yeast[29–35]. In both systems, tRNA-halves have been demonstrated to interact with translational machinery through a wide variety of interacting partners[18], or by directly interacting with the translating ribosome in a codon-specific manner[36], thereby dampening cellular translation in response to stress. Additionally, tDRs have been demonstrated to be required for ribosome biogenesis in human cells, thus promoting translation[37]. Thus, unlike typical RNAi-related small RNA pathways, which decrease target mRNA expression through complementary base pairing, tDRs exhibit highly diverse functions. Due to this functional diversity, their roles in most cellular contexts remain poorly understood.

While tDRs accumulate in eukaryotic cells under conditions of stress, they also accumulate endogenously under normal conditions, especially in the mature sperm of mammals. In typical mammalian tissues, miRNAs comprise the majority of the overall small RNA pool. However, in mammalian sperm, tDRs account for >50% of all small RNAs compared to -10% for miRNAs[5,38]. Further, tDRs in sperm have been demonstrated to be regulated by the paternal environment, modulate post-fertilization embryonic gene expression, and to transmit non-genetically inherited phenotypes to offspring[4–7,39]. The latter has been demonstrated by microinjecting either tDRs or anti-tDRs (antisense RNAs that bind endogenous tDRs and block their activity) into naïve zygotes, which can develop into offspring exhibiting the same non-genetically inherited phenotypes normally transmitted by paternal environmental exposure[4–7]. Because tDRs are extensively chemically modified and these modifications are required for full biological activity, synthetic tDRs generated in vitro often fail to recapitulate endogenous function, yielding variable outcomes in microinjection experiments[4,5]. In contrast, anti-tDRs do not rely on recapitulating these modifications and instead act by competitively inhibiting endogenous tDRs, making them a more robust and reliable approach for probing tDR function in early embryos[5]. Yet, how tDRs function to regulate gene expression and development to transmit inherited information is still unclear. One example of this phenomenon is via an altered paternal diet of male mice (with either low protein or high fat), which leads to an upregulation of a specific tDR in sperm, 5′ tDR-Glycine-GCC. 5′tDR-Glycine-GCC has been shown to regulate the

transcription of genes that have co-opted the Murine Endogenous Retrovirus L (MERVL) as a promoter in both preimplantation embryos and mouse embryonic stem cells (mESCs)[5]. Interestingly, the regulation of MERVL genes during murine zygotic genome activation at the 2-cell stage and in mESCs has been associated with totipotency, suggesting that sperm tDRs can regulate early developmental cell-fate decisions[40]. While the mechanisms underlying how 5′tDR-Glycine-GCC regulates gene expression in embryos is still unclear, it was demonstrated in mESCs to regulate the non-coding small nuclear RNA (snRNA) U7, which subsequently regulates histone RNA levels, chromatin compaction, and expression of MERVL-driven genes[26]. 3′ tDRs were additionally shown to inhibit long terminal repeat retrotransposon mobility in mESCs by interfering with reverse transcription, which canonically requires tRNAs as a primer for synthesis[27].

These findings and others demonstrate that sperm tDRs transmit non-genetically inherited phenotypes to offspring. However, many questions remain, including both (1) how tDRs are generated endogenously within the male reproductive tract and (2) how they function post-fertilization to program the inheritance of non-genetically inherited phenotypes in offspring. RNaseA and T2 enzymes have been implicated in stress-induced tDR biogenesis in cell culture studies, yet it is still unknown if these enzymes generate tDRs in sperm under normal physiological conditions. Finally, it is unclear whether tDR accumulation in sperm is specific to mammals or if this accumulation is broadly conserved among metazoans. Studies in *Drosophila* suggest that tDRs can accumulate in sperm and transmit non-genetic phenotypes to progeny[41], yet the ubiquity of this finding in other phyla has yet to be explored.

Here, we establish *C. elegans* as a model to study tDR biogenesis and regulatory functions, as well as defining the role of tDRs in transmitting non-genetically inherited phenotypes to progeny. We found that tDRs accumulate endogenously in *C. elegans* sperm as they do in mammalian sperm, and that this accumulation is regulated by the RNaseT2 enzyme, RNST-2. Further, we find that tDR accumulation in sperm regulates early embryonic gene expression and that tDR accumulation is associated with the inheritance of phenotypes such as improved survival during L1 starvation and reduced heat shock resistance. Notably, we find that the transmission of these phenotypes is specific to 5′tDRs Gly-GCC and Glu-CTC, as inhibiting the regulatory functions of these tDRs with antisense RNAs blocks the transmission of altered embryonic gene expression and the inheritance of phenotypes. These findings reveal a molecular pathway that regulates tDR biogenesis in the germline and demonstrate that tDR accumulation in sperm, as well as the ability of these small RNAs to communicate non-genetically inherited phenotypes to progeny, is conserved amongst metazoans.

## Results

### tDRs are enriched in *C. elegans* sperm

While tDRs have been demonstrated to accumulate to high levels in mammalian sperm, tDRs are generally understudied in *C. elegans*, particularly in the germline. However, it should be noted that three previous studies have identified tDR accumulation in *C. elegans*, one during adult aging[42], another on specific cleavage of tRNA-Tyrosine-GUA in response to oxidative stress[36], and during cisplatin exposure[43]. These studies were performed on whole adult hermaphrodite worms, thus lacking the ability to assess tDR accumulation in sperm. One reason for the dearth of information on tDRs in worms is that most previous studies specifically purified 18–30 nt RNAs for small RNA cloning and sequencing. In mammals, tDRs (and particularly tRNA-halves) that are abundant in sperm are >30 nt[5,38]; thus, these larger tDRs would not have been captured in most worm small RNA studies. To determine if tDRs are present under normal conditions in *C. elegans*, we isolated adult *fog-2* males and purified sperm. For this and all subsequent experiments, we used the *fog-2(q71)* mutant background to

generate gonochoristic populations of male and female worms that require outcrossed mating for reproduction. The *fog-2* mutation produces hermaphrodites that fail to undergo spermatogenesis and therefore generate only oocytes[44]. In contrast, *fog-2* is neither expressed nor required in males, which undergo normal spermatogenesis and produce wild-type sperm. We then performed small RNA-seq on whole males as well as purified sperm. Importantly, for these experiments, and all others throughout this study, we size-selected 18−40 nt RNA using polyacrylamide gel electrophoresis (PAGE). As expected, and previously demonstrated, we find that the predominant small RNA species present in *C. elegans* males are endo-siRNAs and miRNAs. However, cloning 18−40 nt RNAs reveals reads mapping to tRNAs, representing tDRs, at 3.3% of all reads in whole males. Strikingly, this proportion increases to 13.8% in purified sperm, which is nearly twice the proportion of miRNAs in sperm (Fig. 1A, B−*P* < **0.05**). Thus, tDRs are enriched in *C. elegans* sperm as they are in mammalian sperm. Interestingly, specific tDRs such as tDR-Gly-GCC and tDR-Glu-CTC accumulate in *C. elegans* sperm (Fig. 1A), analogous to their relative abundance in mammalian sperm. As previously reported, we additionally find that mRNA-fragments are also enriched in *C. elegans* sperm, which are likely the remnants of spermatogenic mRNA degradation that occurs as sperm undergo final steps of maturation in most animals[45,46].

tDRs have been observed in *C. elegans*, but their molecular features have not been described. When plotting the length of small RNAs within our male and sperm small RNA data, we find the expected accumulation of 22G-RNAs (22 nt endo-siRNAs that predominantly begin with a G), and 26G-RNAs (26 nt endo-siRNAs that predominantly begin with a G). However, we also noticed that sperm contain abundant small RNAs that are >30 nt long (Supplementary Fig. 1A). We then plotted the size distribution and first nucleotide of tRNA mapping reads. tDRs range from 18 to 40 nt in both whole males and purified sperm. Interestingly, there is a noticeable accumulation of 36 nt tDRs, which correspond to the size of tRNA-halves in both males and sperm (Fig. 1C and Supplementary Fig. 1C). Further, tDRs of specific lengths and starting nucleotides accumulate in both males and sperm (Fig. 1C), unlike rRNA-fragments, which skew towards smaller RNAs with equal distribution of starting nucleotide, indicating the former are stable species rather than random degradation products (Supplementary Fig. 1B).

Using standard ligation-based small RNA-seq protocols (without any RNA treatment), tDRs in mammalian sperm are sequenced predominantly as 5′ fragments of tRNAs (>80%)[5,38,47]. However, this bias towards 5′tDRs has been determined to result from molecular features of tDRs, such as 3′ chemical modifications and modified RNA bases, which prevent conventional approaches from cloning every tDR species, particularly 3′ tDRs. Indeed, new small RNA-seq techniques that can clone RNAs irrespective of chemical modifications at internal sites and at termini have revealed that both 5′ and 3′ tDRs accumulate equally in mammalian sperm[48–50]. Interestingly, using a standard ligation-dependent small RNA-seq approach on our *C. elegans* whole male and purified sperm samples, we find a near equal distribution (52 vs. 48%) of reads mapping to the 5′ and 3′ regions of tRNAs, respectively, in whole males, with a slight skew in sperm (42 vs. 58%) (Fig. 1D). This suggests that chemical modifications present in mammalian sperm that drive the extreme skew in cloning efficiency (>80%) are not present in *C. elegans* sperm.

Interestingly, specific tRNA isoacceptors produce specific types of tDRs. We quantitated the distribution of small RNA-seq read depth across the mature tRNAs Gly-GCC, Glu-CTC, and Ser-AGA. For both Gly-GCC and Glu-CTC, we find a specific accumulation of 5′tDRs in both males and sperm; conversely, for Ser-AGA, 3′ tDRs accumulate and are enriched in sperm (Fig. 1E and Supplementary Fig. 1D). Interestingly, individual tRNA isoacceptors display a wide range of 5′ and 3′ fragment biases, suggesting isoacceptor-specific differences in tDR

biogenesis and stability in sperm (Supplementary Fig. 1E). Overall, these findings demonstrate that tDRs are present in *C. elegans* and are enriched in sperm as they are in mammalian sperm. Further, specific tDR products from both tRNA 5′ and 3′ portions accumulate from different isoacceptors, including tDRs such as 5′ Gly-GCC, 5′ Gly-CTC, and Gln-TTG, which have been demonstrated to function to regulate embryogenic gene expression and development in mammals[5,26,51,52]. Thus, the accumulation of specific tDRs appears to be a conserved hallmark of metazoan sperm, underlying the importance of studying the functions of tDRs in fertility and non-genetic inheritance.

## tDR abundance in males is regulated by *rnst-2*

RnaseA and RnaseT2 enzymes have been implicated in tDR biogenesis and specifically found to cleave full-length tRNAs into tRNA-halves in mammalian cell culture and yeast models. However, the nucleases required to generate tDRs in the mammalian germline have yet to be identified. The *C. elegans* genome encodes no RnaseA orthologs and only a single RnaseT2 ortholog, *rnst-2*. To determine if *rnst-2* is required to generate tDRs in worms, we utilized CRISPR-Cas9 editing to generate a mutant containing a small indel towards the 5′ end of the *rnst-2* gene, resulting in a frameshift in its coding sequence and a premature STOP at amino acid 15. Additionally, using CRISPR and homologous recombination we generated a second mutant with a point mutation predicted, from homology with the human protein which has been analyzed both structurally and catalytically[53], to affect the catalytic activity of the enzyme by introducing a Histidine to Alanine substitution at amino acid 60 (Supplementary Fig. 2A). We then crossed these mutants (deletion (Δ) − *pen1* and catalytic mutant (*cat*) − *pen4*) into the *fog-2* background, isolated *rnst-2*[+/−] heterozygous (balanced by *tmc16*) and *rnst-2*[−/−] homozygous males for each genotype, purified RNA from whole males, and then performed 18−40 nt small RNA-seq on these samples. Importantly, homozygous deletion mutants show a nearly complete loss of *rnst-2* expression relative to heterozygous males, while the catalytic point mutations retain partial expression of the gene (Supplementary Fig. 2B). Surprisingly, we find a significant enrichment of tRNA mapping reads representing tDRs in both *rnst-2*[Δ/Δ] and *rnst-2*[cat/cat] males compared to heterozygotes (Δ/Δ 8.6% vs. Δ/+ 3.5% and *cat/cat* 8.1% vs. *cat*/+ 3.7%), driven primarily by elevated abundance of a subset of tRNA isoacceptors, including Gly-GCC and Glu-CTC (Fig. 2A, B and Supplementary Fig. 2E). Further we find a striking correlation in the accumulation of tDRs, rRNA-fragments, and mRNAs-fragments in both *rnst-2*[Δ/Δ] and *rnst-2*[cat/cat] males compared to heterozygous controls, suggesting similar functional consequences on these RNA species in distinct *rnst-2* mutations (Supplementary Fig. 2D).

Because tDR Gly-GCC and Glu-CTC accumulate as 5′tDRs in our male and sperm small RNA dataset, we next assessed the distribution of reads across tRNA sequences. We found that both *rnst-2* mutations exhibited a significantly higher fraction of 5′tDRs overall (Δ/Δ 86.0% vs. Δ/+ 62.3% and *cat/cat* 82.3% vs. *cat*/+ 59.4%) (Fig. 2C). Additionally, plotting the length and first nucleotide distribution of tRNA mapping reads revealed a significant increase in >30 nt fragments representing tRNA-halves, predominantly in 31−36 nt species (Fig. 2D and Supplementary Fig. 2G). While rRNA-fragments also increase markedly in both *rnst-2* mutants (Fig. 2A, B and Supplementary Fig. 2E), the distribution of fragments is collectively driven towards smaller RNAs, with a random first nucleotide representation (Supplementary Fig. 2F), suggesting that RNST-2 appears to eliminate degradation products generated from rRNAs. However, more notably, RNST-2 specifically degrades tRNA-halves into smaller RNAs or entirely, revealed by the increase in overall abundance and in longer tDRs present in *rnst-2* mutant males (Fig. 2D, E and Supplementary Fig. 2G).

Focusing on tDRs generated from specific tRNA-isoacceptors, we again quantitated the nucleotide read depth across specific mature tRNA sequences from our *rnst-2* mutant male RNA-seq. For the 5′tDRs

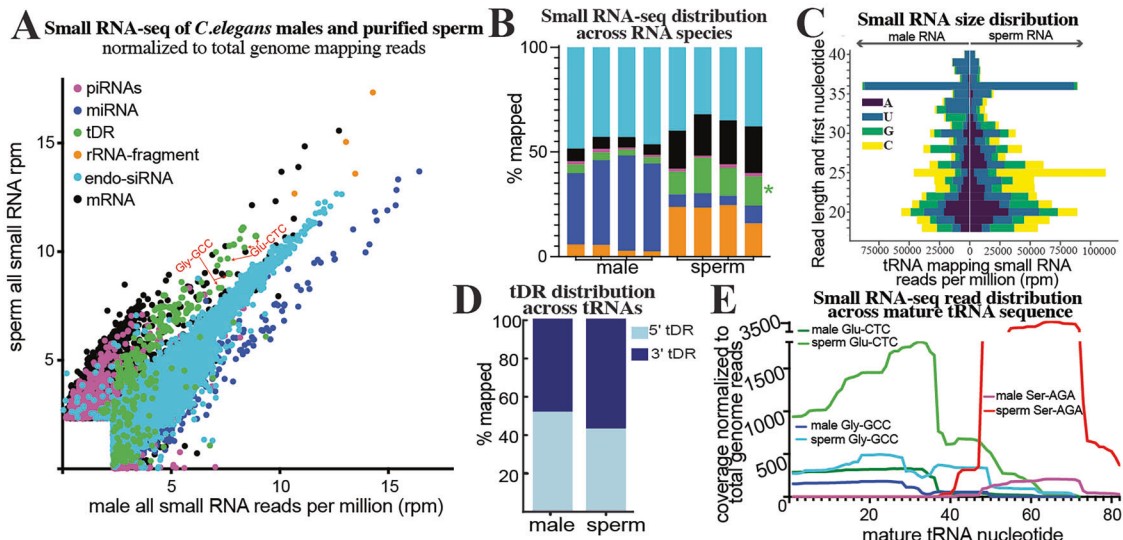

**Fig. 1 | Small RNA-seq reveals tDRs are enriched in C. elegans sperm.** Small RNA-seq was performed on PAGE-purified 18–40 nt RNA from *C. elegans* whole males and purified sperm. **A** small RNA reads were aligned to specific small RNA classes (rRNAs, tRNAs, miRNAs, and piRNAs), as well as to the transcriptome (mRNA-fragments and endo-siRNAs), and quantitated. Quantitated counts were then normalized to total genome mapping reads as reads per million (rpm). For each RNA, the normalized rpm was plotted in a scatter plot. Different small RNA classes are colored according to the legend. **B** Total small RNA reads mapping to each class of RNAs quantitated as a percentage of total genome mapping reads as stacked bar graphs for each replicate of the experiment ($n = 4$). Color legend is the same as in (**A**). Total small RNA reads mapping to each RNA class were compared between conditions using two-sided Mann–Whitney *U* tests with Benjamini–Hochberg false discovery rate (FDR) correction; asterisks indicate FDR-adjusted $P < 0.05$. **C** Reads mapping to tRNAs were quantitated for RNA length (*y*-axis) and plotted as their starting nucleotide in a diverging bar-graph (*x*-axis–total quantitated reads). The first nucleotide is indicated by color in the legend. Read-length distributions were compared between conditions using two-sided Mann–Whitney U tests with Benjamini–Hochberg false discovery rate (FDR) correction; asterisks indicate bins with FDR-adjusted $P < 0.05$. **D** tRNA mapping reads were quantitated for mapping specifically to the 5′ or 3′ portions (mature tRNA separated by anticodon) and plotted as the percentage mapping to each. Proportions of 5′ and 3′ tRNA-derived fragments were compared between males and sperm using two-sided Welch's t-tests on replicate-level values ($P = 0.046$). **E** Reads mapping to specific tRNA iso-acceptors of Glu-CTC, Gly-GCC, and Ser-AGA were quantitated as read depth (*y*-axis −number of times each nucleotide mapped) across the mature tRNA sequence (*x*-axis). Nucleotide-level read depth across the mature tRNA sequence was quantified for Glu-CTC, Gly-GCC, and Ser-AGA and compared between male and sperm samples using two-sided Welch's t-tests, with significant differences observed for Glu-CTC ($P = 3.57 \times 10^{-40}$), Gly-GCC ($P = 6.46 \times 10^{-12}$), and Ser-AGA ($P = 9.66 \times 10^{-25}$). Source data are provided as a Source data file.

Gly-GCC and Glu-CTC, we find a prominent overall increase in levels in *rnst-2* homozygous mutants compared to controls, which is supported by northern blot analysis of *rnst-2*$^{cat/cat}$ and WT male RNA (Fig. 2E, F). Conversely, we find no significant changes in the predominately 3′ tDR Ser-AGA. We conclude that >30 nt 5′tDRs accumulate in *rnst-2* mutants, suggesting that RNST-2 normally functions to reduce the levels of these RNAs. Further, elimination of 5′tDR Gly-GCC and Glu-CTC halves is dependent on a predicted catalytic amino acid present in RNST-2. Thus, while RNST-2 is not the enzyme required to initially cut tRNAs into tRNA-halves, our findings suggest that it regulates the processing of tRNA-halves into smaller tDRs or eliminates them altogether.

## *rnst-2* regulates tDRs specifically in sperm and not in females

To determine where *rnst-2* functions to regulate tDR accumulation, we endogenously tagged *rnst-2* N-terminally with GFP using CRISPR and homologous donor recombination. We then crossed *gfp::rnst-2* into the *fog-2* background to determine where RNST-2 is expressed. Importantly, *gfp::rnst-2* males display normal male fertility, suggesting that the GFP::RNST-2 fusion is indistinguishable from wild type (Supplementary Fig. 2C). As predicted from the accumulation of tDRs in both *rnst-2*$^{Δ/Δ}$ and *rnst-2*$^{cat/cat}$ males, we find GFP::RNST-2 expression specifically in sperm (Fig. 3A). While we cannot rule out low levels of expression elsewhere, sperm is the only place we identify detectable GFP signal in adult males and females.

We next aimed to determine if loss of *rnst-2* and the subsequent changes to tDR abundance in sperm lead to male fertility phenotypes. To assess fertility and determine sex-specific influences on reproduction, we quantitated the brood size at 20 °C of *rnst-2* mutant males

(both Δ and *cat* mutants) crossed with *fog-2* (WT) females, as well as reciprocal crosses of *fog-2* (WT) males with mutant females. We find that *rnst-2* homozygous males crossed with WT *fog-2* females produce a significantly decreased overall brood size of ~200 progeny for both Δ and *cat* mutants. As controls, we quantitated the broods of WT males crossed with WT females as well as *rnst-2*$^{Δ/+}$ heterozygous males crossed with WT females, which each produce broods of ~300 progeny. Further, we find that *rnst-2*$^{Δ/+}$ and *rnst-2* mutant females mated with WT males produce normal broods of ~300 progeny (Supplementary Fig. 2C). These findings demonstrate that *rnst-2* is required for normal male fertility in *C. elegans*.

To determine if tDRs regulated by *rnst-2* specifically accumulate in sperm, we isolated males and purified sperm from *rnst-2*$^{cat/cat}$ mutants for small RNA-seq. We then quantitated small RNAs compared to our WT *fog-2* sperm small RNA data (Fig. 1). We find that specific tDRs, including Gly-GCC and Glu-CTC, are enriched in *rnst-2*$^{cat/cat}$ sperm (Fig. 3B–D). This enrichment is obscured when small RNA data are normalized to *total genome mapping reads*. The standard reads per million (rpm) normalization metric is heavily influenced by the large increase in rRNA-derived degradation products in *rnst-2* mutants, thereby compressing the dynamic range and masking enrichment of other small RNA subtypes (Supplementary Fig. 3A, B). To account for this accumulation, we normalized our data to *genome mapping reads minus rRNA mapping reads* to account for differential rRNA-fragment accumulation in *rnst-2*$^{cat/cat}$ sperm compared to WT, revealing the relative contribution of tRNA-derived reads was significantly increased in *rnst-2*$^{cat/cat}$ sperm compared to WT (Fig. 3D), with Gly-GCC tDRs and to a lesser extent Glu-CTC tDRs driving much of this shift (Fig. 3B).

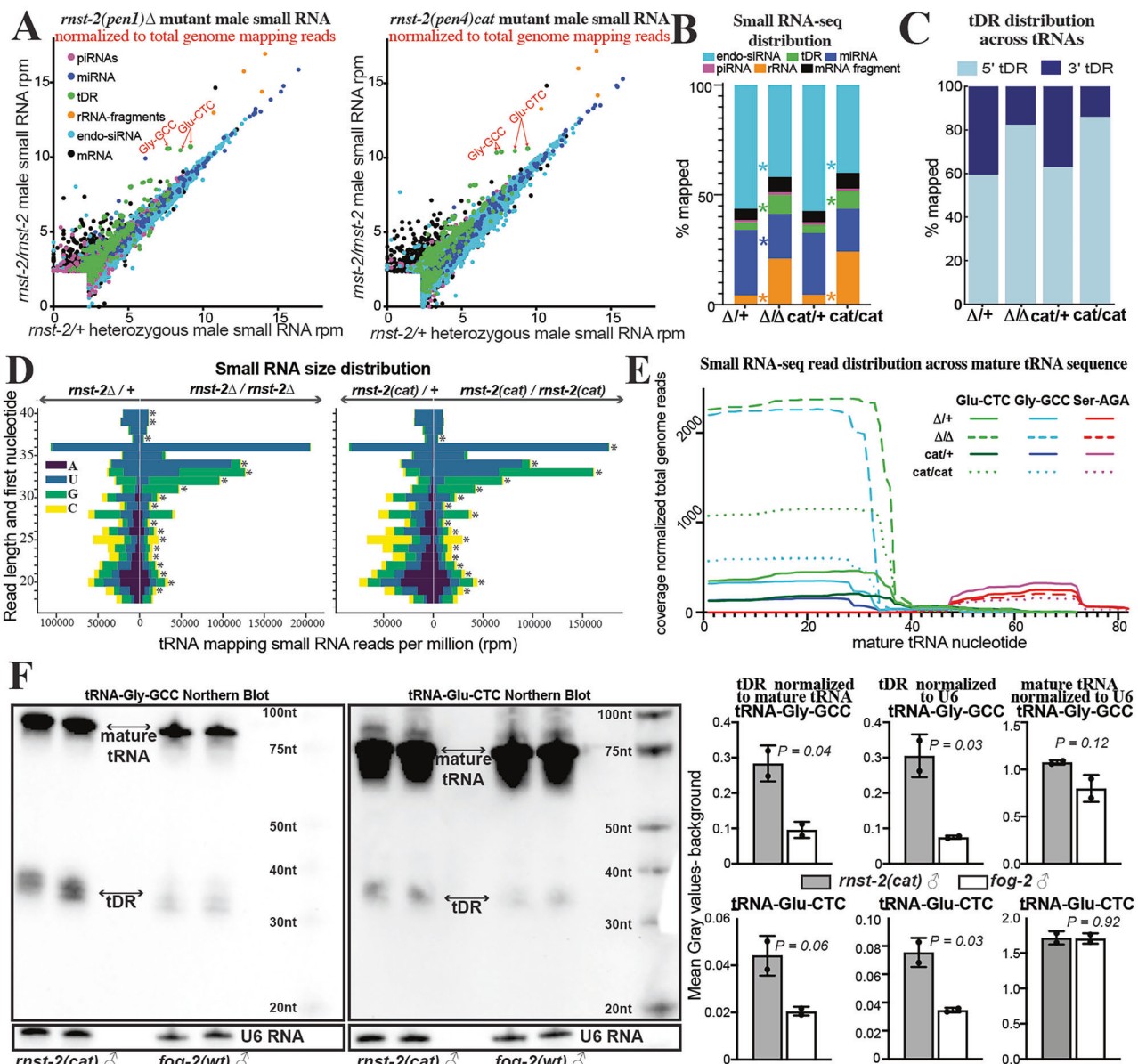

**Fig. 2 | *rnst-2* regulates tDR accumulation and length.** Small RNA-seq was performed on *C. elegans* whole male *rnst-2* mutant (*pen1 – Δ* and *pen4 – catalytic*) as well as heterozygous controls. **A** Small RNA reads were aligned to specific small RNA classes (rRNAs, tRNAs, miRNAs, and piRNAs), as well as to the transcriptome (mRNA fragments and endo-siRNAs), and quantitated. Quantitated counts were then normalized to total genome mapping reads as reads per million (rpm). For each RNA, the normalized rpm was plotted in a scatter plot. Different small RNA classes are colored according to the legend. **B** Total small RNA reads mapping to each class of RNAs quantitated as a percentage of total genome mapping reads as stacked bar graphs of the average of biological replicates (*n* = 4–5). Total small RNA reads mapping to each RNA class were compared between conditions using two-sided Mann–Whitney *U* tests with Benjamini–Hochberg false discovery rate (FDR) correction; asterisks indicate FDR-adjusted *P* < 0.05. **C** tRNA mapping reads were quantitated for mapping specifically to the 5′ or 3′ portions (mature tRNA separated by anticodon) and plotted as the percentage mapping to each. Proportions of 5′ and 3′ tRNA-derived fragments were compared between *rnst-2* mutant homozygotes and heterozygotes and between *rnst-2* catalytic mutant homozygotes and heterozygotes using two-sided Welch's t-tests on replicate-level values (*P* = 7.93 × 10⁻⁸ and *P* = 0.022, respectively). **D** Reads mapping to tRNAs were quantitated for RNA length (*y*-axis) and plotted as their starting nucleotide (legend) in a diverging bar-graph (*x*-axis−total quantitated reads). Read-length distributions

were compared between conditions using two-sided Mann–Whitney *U* tests with Benjamini–Hochberg false discovery rate (FDR) correction; asterisks indicate bins with FDR-adjusted *P* < 0.05. **E** Reads mapping to specific tRNA isoacceptors of Glu-CTC, Gly-GCC, and Ser-AGA were quantitated as read depth (*y*-axis−number of times each nucleotide mapped) across the mature tRNA sequence (*x*-axis) for each homozygous *rnst-2* mutant and heterozygous control. Nucleotide-level read depth across the mature tRNA sequence was quantified for Glu-CTC, Gly-GCC, and Ser-AGA and compared between *rnst-2* mutant homozygotes and heterozygotes and between *rnst-2* catalytic mutant homozygotes and heterozygotes using two-sided Welch's t-tests. Glu-CTC and Gly-GCC showed significant differences in both comparisons (Glu-CTC: Δ/+ vs. Δ/Δ *P* = 2.89 × 10⁻³⁰, *cat/+* vs *cat/cat P* = 2.08 × 10⁻³⁹; Gly-GCC: Δ/+ vs. Δ/Δ *P* = 1.80 × 10⁻³³, *cat/+* vs *cat/cat P* = 4.02 × 10⁻³⁵), whereas Ser-AGA did not differ significantly (Δ/+ vs. Δ/Δ *P* = 0.91; *cat/+* vs *cat/cat P* = 0.92). **F** Levels of Gly-GCC and Glu-CTC mature tRNAs and tDRs were determined by northern blot analysis of total RNA from adult male worms of the indicated genotypes using digoxigenin-labeled DNA oligonucleotide probes specific to Gly-GCC or Glu-CTC. U6 snRNA was analyzed as a loading control. Quantifications of tDR levels normalized to the corresponding mature tRNA or to U6, and of mature tRNA levels normalized to U6, are shown in the accompanying quantification panels. *n* = 2 independent biological replicates; data are presented as mean ± SEM (unpaired two-tailed *t*-test). Source data are provided as a Source data file.

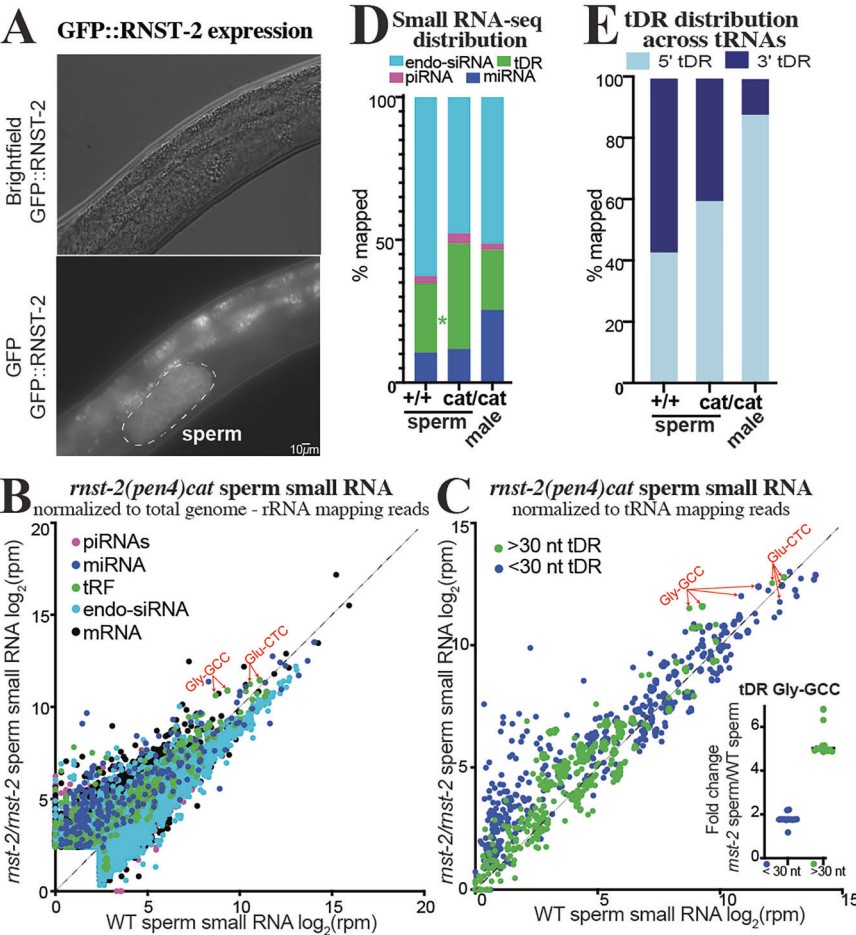

**Fig. 3 | rnst-2 regulates tDR accumulation in sperm. A** *rnst-2* endogenously tagged at its N-termini with GFP is expressed in *C. elegans* sperm (outlined in dotted line). Signal outside of the sperm is gut auto-fluorescence. This pattern was observed in 3 independent cohorts of worms. **B**–**E** Small RNA-seq was performed on *rnst-2(cat)* sperm and whole males and compared to WT sperm small RNA data. **B** Scatter plot of small RNA-seq data normalized to total genome minus rRNA mapping reads as reads per million (rpm) from WT sperm (*x*-axis–also Fig. 1) and *rnst-2^cat/cat^* mutant sperm (*y*-axis). **C** tRNA mapping reads were separated as >30 or <30 nucleotides in length and quantitated, as reads per million (rpm) tRNA mapping reads, for all tRNA isoacceptor genes. Each dot represents a quantitated tRNA with quantitated reads mapping to that tRNA >30 nucleotide in green and <30 in blue, thus each tRNA isodecoder is represented as 2 dots. The inset graphs show a dot-plot of each tRNA-Gly-GCC isodecoder quantitation. *n* = 4 independent

biological replicates per genotype, horizontal lines indicate medians. **D** Distribution of small RNAs mapped to endo-siRNAs, piRNAs, miRNAs, and tDRs in WT and *rnst-2^cat/cat^* mutant sperm and males. Data presented is the average of biological replicates (*n* = 3–4). Total small RNA reads mapping to each RNA class were compared between conditions using two-sided Mann–Whitney *U* tests with Benjamini–Hochberg false discovery rate (FDR) correction; asterisks indicate FDR-adjusted *P* < 0.05. **E** Distribution of tRNA mapping reads to the 5′ or 3′ portions of mature tRNA and plotted as the average across replicates as the percentage of mapped reads. Proportions of 5′ and 3′ tRNA-derived fragments were compared between WT sperm and rnst-2 catalytic mutant sperm using two-sided Welch's *t*-tests on replicate-level values (*P* = 8.93 × 10⁻⁴). Source data are provided as a Source data file.

To determine if *rnst-2* regulates tDR length in sperm, we plotted the length and first nucleotide of tRNA mapping reads for *rnst-2^cat/cat^* and WT sperm. Similar to the trend observed with *rnst-2* mutant males, we find that tDR length increases overall, with a particular accumulation of 32–35 nt species (Supplementary Fig. 3C). To assess if the accumulation of larger tDRs is specific to any specific tRNA isoacceptors, we quantitated >30 and <30 nt small RNAs that map to tDRs. This analysis revealed that >30 nt tDRs (indicative of tRNA-halves), particularly for tDR-Gly-GCC, are enriched in mutant compared to WT sperm (Fig. 3C). Interestingly, shorter <30 nt tDRs for Gly-GCC are less enriched in *rnst-2^cat/cat^* sperm compared to WT (>30 nt -5-fold vs. <30 nt -2-fold increase), indicating a shift in tDR length dependent on *rnst-2* in sperm. Consistent with increased accumulation of >30 nt tRNA-halves, *rnst-2* catalytic mutant sperm exhibited a significantly higher fraction of tDRs mapping to the 5′ arm of tRNAs compared to WT sperm (Fig. 3E). These results demonstrate that *rnst-2*, and specifically the catalytic residue of the enzyme, regulates the

accumulation of >30 nt 5′tDRs in sperm, which closely resemble functional sperm tDRs in mammals.

Our brood analysis of *rnst-2* mutants (Supplementary Fig. 2C) revealed that both null (*Δ*) and catalytic (*cat*) male mutants have a specific decrease in the number of sired progeny, while female mutants are unaffected in their fertility. This raised the question of whether tDRs are also present in *C. elegans* females, and further, if they are regulated by *rnst-2*. To address these questions, we isolated *rnst-2^Δ/+^* heterozygous and *rnst-2^Δ/Δ^* homozygous adult females in the *fog-2* background. To specifically isolate the small RNA profile of the female germline and soma (without any contribution from developing embryos), we picked virgin L4 animals and allowed them to develop to adults for 24 h in the absence of males. We then performed 18–40 nt small RNA-seq on these samples to quantitate their small RNA profiles. We find that tDRs are less abundant in females overall compared to males (2.2% in female *rnst-2^Δ/+^* compared to 3.5% in male *rnst-2^Δ/+^*) and that tDRs levels increase very little, and without statistical significance,

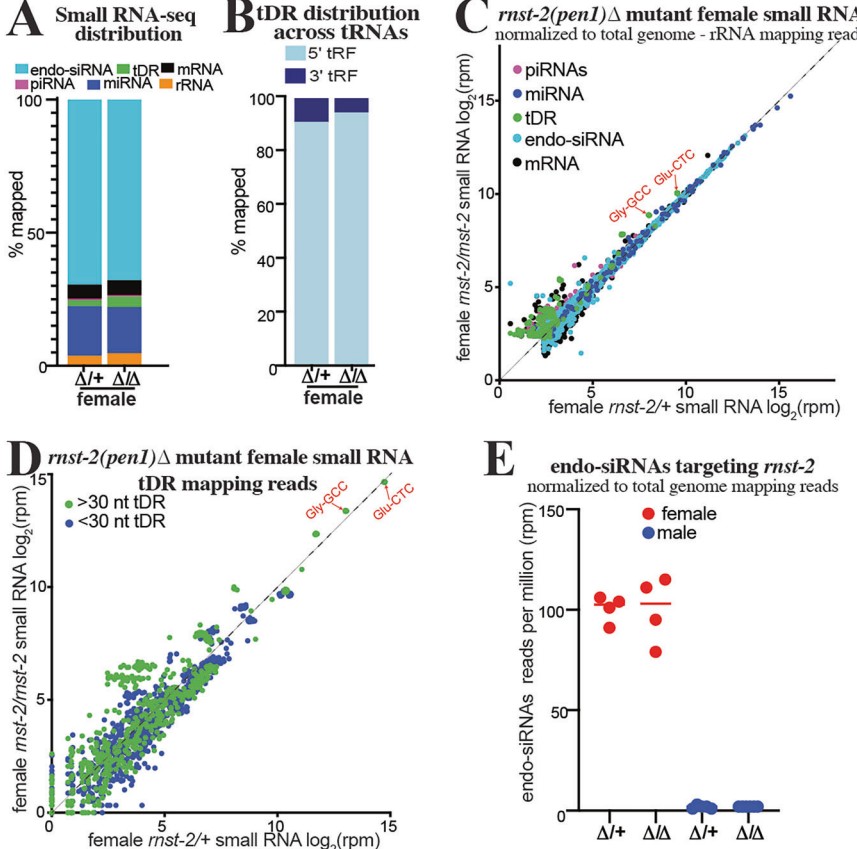

**Fig. 4 | *rnst-2* dependent regulation of tDRs is suppressed in females.** Small RNA-seq was performed on *C. elegans* whole female (without embryos) *rnst-2*$^{\Delta/\Delta}$ mutants as well as heterozygous controls. **A** Distribution of small RNAs mapped to small RNA classes in *rnst-2*$^{\Delta/\Delta}$ and *rnst-2*$^{\Delta/+}$ females. Total small RNA reads mapping to each RNA class were compared between conditions using two-sided Mann–Whitney U tests with Benjamini–Hochberg false discovery rate (FDR) correction; asterisks indicate FDR-adjusted *P* < 0.05. **B** Average distribution of tRNA mapping reads to the 5′ or 3′ portions of mature tRNA plotted as the percentages of mapped reads. Proportions of 5′ and 3′ tRNA-derived fragments were compared between rnst-2Δ/Δ females and rnst-2Δ/+ heterozygotes using two-sided Welch's *t*-tests on replicate-

level values and did not differ significantly. **C** Scatter plot of small RNA-seq data normalized to total genome minus rRNA mapping reads as reads per million (rpm) from *rnst-2*$^{\Delta/+}$ (*x*-axis) and *rnst-2*$^{\Delta/\Delta}$ females (*y*-axis). **D** tRNA mapping reads were separated as >30 or <30 nucleotides in length and quantitated, as reads per million (rpm), for all tRNA isodecoders. *rnst-2*$^{\Delta/+}$ (*x*-axis) and *rnst-2*$^{\Delta/\Delta}$ females (*y*-axis). **E** endo-siRNAs (antisense mRNA mapping) reads normalized to total genome mapping reads were quantitated as reads per million from *rnst-2*$^{\Delta/\Delta}$ and *rnst-2*$^{\Delta/+}$ female and male (Fig. 2) small RNA data. *n* = 4 independent biological replicates per genotype, horizontal lines indicate medians. Source data are provided as a Source data file.

in *rnst-2*$^{\Delta/\Delta}$ compared to heterozygous females (Fig. 4A–C and Supplementary Fig. 4A, B), in contrast to our findings in mutant males (Fig. 2). Further, we find that that the overall length and first nucleotide distribution of tDRs in *rnst-2*$^{\Delta/\Delta}$ and *rnst-2*$^{\Delta/+}$ females is similar, with long >30 nt tDRs dominating (Supplementary Fig. 4C), again unlike males with a functional copy of *rnst-2* (*rnst-2*$^{\Delta/+}$) where small tDRs preferentially accumulate. Further, we found that tDR Gly-GCC and Glu-CTC increase only subtly in *rnst-2*$^{\Delta/\Delta}$ females compared to *rnst-2*$^{\Delta/+}$ heterozygotes (Fig. 4C and Supplementary Fig. 4B). This finding additionally extends to >30 nt tDRs, which do not increase in *rnst-2* mutant females as they do in males (Fig. 4D). Interestingly, analysis of endo-siRNAs (small RNAs mapping antisense to mRNAs) from *rnst-2*$^{\Delta/\Delta}$ and *rnst-2*$^{\Delta/+}$ females revealed abundant small RNAs targeting *rnst-2* in both data sets. These endo-siRNAs targeting *rnst-2* are only present in females and not males, indicating that they could be functioning to silence the expression of *rnst-2* (Fig. 4E). Additional analysis revealed that these endo-siRNAs are 22G-RNAs, further suggesting that they could repress *rnst-2* expression in the germline by the endo-siRNA pathway (Supplementary Fig. 4D). These findings, paired with the lack of a *rnst-2* female fertility phenotype and undetectable GFP::RNST-2 expression in females, suggest that the absence of *rnst-2* precludes processing of >30 tDRs or tRNA-halves into smaller fragments.

## *rnst-2* mutant males transmit non-genetically inherited phenotypes to their offspring

In mammals, tDRs, particularly tDR-Gly-GCC, were shown to be regulated by paternal diet and to transmit non-genetically inherited phenotypes to offspring, including altered embryonic gene expression and adult metabolism[4–6]. Since tDRs accumulate substantially in the sperm of *rnst-2*$^{\Delta/\Delta}$ animals, mutant males provide a potential model to determine if tDRs can transmit non-genetically inherited phenotypes to progeny in *C. elegans*. To establish worms as a model to study the functions of tDRs in intergenerational non-genetic inheritance, we mated *rnst-2*$^{\Delta/\Delta}$ males and WT *fog-2* males, as controls, with WT *fog-2* females to produce progeny sired by sperm with increased and normal levels of tDRs (Fig. 5A). To determine if *rnst-2*$^{\Delta/\Delta}$ mutant sperm produce embryos with altered gene expression compared to control sperm, we performed single-embryo RNA-seq on 2-cell and 8-cell embryos (representing embryonic stages pre- and post-zygotic genome activation (ZGA)). Single-embryo RNA-seq on worm embryos allowed us to quantitate the gene expression of >6000 genes across 4-orders of magnitude of expression. Using this approach, we compared the gene expression of *rnst-2*$^{\Delta/\Delta}$ male sired to WT male sired embryos to determine if increased delivery of sperm tDRs influences post-fertilization gene expression. For 2-cell embryos, we find subtle changes in gene

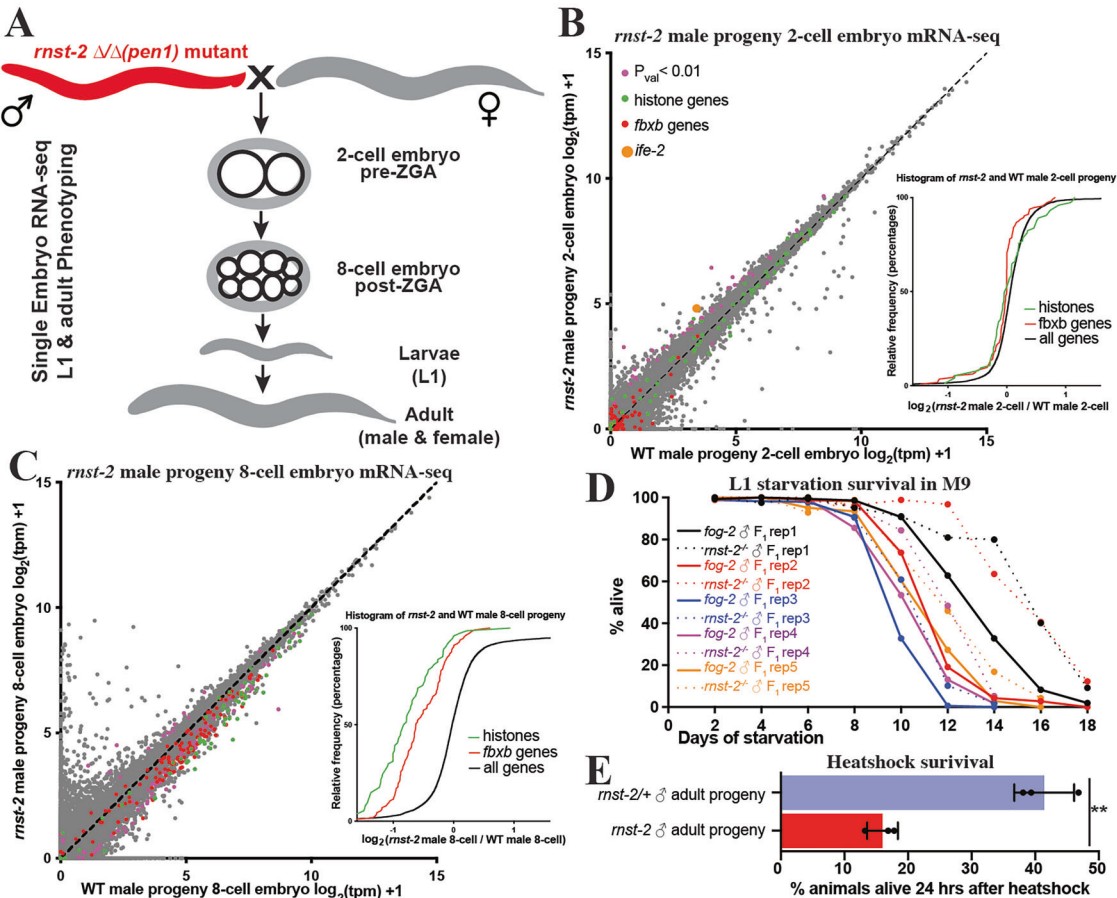

**Fig. 5 | *rnst-2* mutant males transmit non-genetically inherited phenotypes to offspring. A** Schematic of the mating scheme and developmental stages of offspring phenotypically profiled. **B, C** Single-embryo mRNA-seq was performed on embryos resulting from *rnst-2*$^{Δ/Δ}$ male crossed with WT (*fog-2*) females compared to WT males crossed with WT females. Transcripts per million (tpm) normalized mRNA-seq data represented as scatter plots –WT (*fog-2* X *fog-2*) embryos (*x*-axis) versus *rnst-2*$^{Δ/Δ}$ male X WT female embryos (*y*-axis). Genes with *P* < 0.01 (purple), histone genes (green), *fbxb* genes (red), *ife-2* (orange). The inset graphs represent cumulative distribution frequency plots of all genes, histone genes, and *fbxb* genes. **B** 2-cell embryo pre-zygotic genome activation (*n* = 30 WT, *n* = 25 *rnst-2*$^{Δ/Δ}$ sired). The inset graphs the cumulative distribution frequency for histone genes, *fbxb* genes, and all remaining genes. **C** 8-cell embryo post-zygotic genome activation (*n* = 20 WT, *n* = 26 *rnst-2*$^{Δ/Δ}$ sired). The inset graphs the cumulative distribution frequency for histone genes, *fbxb* genes, and all remaining genes. **D** The L1 progeny of *rnst-2*$^{Δ/Δ}$ or WT *fog-2* males mated with WT females were assessed for starvation survival over a time course in M9 medium. L1 survival was scored across a time course, with three technical replicates counted per condition every other day. The experiment was biologically replicated five times from independent cohorts of worms. **E** The adult progeny of *rnst-2*$^{Δ/Δ}$ or *rnst-2*$^{Δ/+}$ mated with WT females mated were evaluated for survival after heat-shock. *n* = 3 independent biological replicates; data are presented as mean ± SEM (unpaired two-tailed *t*-test). Source data are provided as a Source data file.

expression in embryos sired by *rnst-2* mutants compared to WT males, as expected from a stage where cells are not yet undergoing active transcription. We identified 38 significantly altered genes (*P* < 0.01), including 2 that were downregulated and 36 that were upregulated (Fig. 5B). Among these, *ife-2*, which encodes an eIF4E translation initiation factor, was the most significantly affected gene and was upregulated 2.5-fold (*P* < 0.5 × 10$^{-5}$ – Fig. 5B and Supplementary Fig. 5A). Interestingly, tDRs have been demonstrated to directly interact with eIF4E proteins under conditions of stress in mammalian cell-culture studies[54,55]. At the 8-cell stage, we find many more differentially expressed genes in embryos fathered by *rnst-2* mutant males compared to WT males, with 106 downregulated and 16 upregulated. Interestingly, we find many histone genes to be downregulated (33 total), and when we assess the expression of all histone genes expressed in 8-cell embryos as a group, we find a decrease in expression overall (Fig. 5C). Histone genes downregulated in embryos sired by *rnst-2* mutant fathers represent all canonical histones, H3, H4, H2A and H2B, but do not include histone variant H3.3 genes

(Supplementary Fig. 5B). Additionally, we also found several *fbxb* genes downregulated, which is an expanded gene class in *C. elegans* which encode for F-box proteins involved in protein ubiquitination and homeostasis. Similar to histone genes, when we assess the expression of all *fbxb* genes expressed in 8-cell embryos, we find that they are coordinately downregulated (Fig. 5C). Assessing *ife-2* regulation in 8-cell embryos as well as histone and *fbxb* genes in 2-cell embryos, we did not find the same coordinated regulation, indicating independent regulatory mechanisms pre- and post-ZGA (Fig. 5B, C and Supplementary Fig. 5A, B). These findings demonstrate that *rnst-2* mutant sperm produce embryos with increased expression of a translation initiation factor at the 2-cell stage pre-ZGA, followed by decreased expression of histone and *fbxb* genes at the 8-cell stage post-ZGA.

We next assessed whether the *rnst-2* mutant male sired F1 progeny inherit any quantifiable phenotypes. Interestingly, we found that *rnst-2* male sired L1 stage F1 progeny survive L1 starvation significantly better than WT male sired progeny across five independent cohorts of

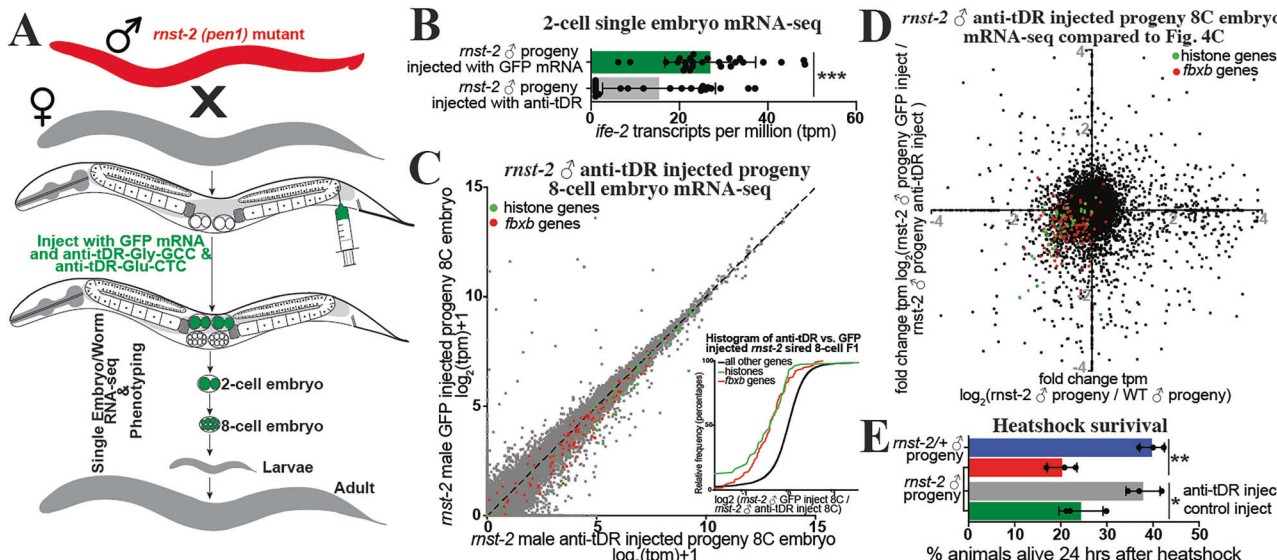

**Fig. 6 | Specific tDRs contribute to the transmission of non-genetically inherited phenotypes to progeny. A** Schematic of our microinjection approach to interfere with specific tDR functions post-fertilization to assess the causality of tDRs in transmitting specific phenotypes to progeny. **B, C** Single-embryo mRNA-seq was performed on 2-cell and 8-cell embryos sired by *rnst-2*$^{\Delta/\Delta}$ males mated with WT *fog-2* females and then microinjected with either anti-tDR (anti-Gly-GCC or anti-Glut-CTC) + GFP mRNA or GFP mRNA alone as a microinjection control group. **B** Bar-plot representing *ife-2* expression in transcripts per million (tpm) for each replicate in our 2-cell single-embryo RNA-seq dataset (***$P = 0.0003$–unpaired two-tailed *t*-test). **C** tpm normalized mRNA-seq data represented as scatter plots–8-cell embryo stage progeny of *rnst-2*$^{\Delta/\Delta}$ males mated with WT females injected with anti-tDR cocktail (*x*-axis) and 8-cell embryo stage progeny of *rnst-2*$^{\Delta/\Delta}$ mated with WT females injected with GFP mRNA control cocktail (*y*-axis). The inset graph represents cumulative distribution frequency plots of all genes, histone genes (colored in

green–$P < 0.0001$ Kolmogorov-Smirnov test), and *fbxb* genes (colored in red–$P < 0.0001$). $n = 18$ GFP injected, $n = 20$ anti-tDR injected. **D** Comparison of single-embryo RNA-seq gene expression fold changes in 8-cell embryo progeny of *rnst-2*$^{\Delta/\Delta}$ males mated to WT females compared to WT 8-cell embryos (*x*-axis–from Fig. 5) and 8-cell embryo progeny of *rnst-2*$^{\Delta/\Delta}$ male mated with WT female microinjected with GFP mRNA compared to 8-cell embryo progeny of *rnst-2*$^{\Delta/\Delta}$ male mated with WT female microinjected with anti-tDR. Dots in the lower left and upper right quadrants represent genes with gene expression rescued by interfering with tDR functions (histone genes–green, *fbxb* genes–blue). **E** The adult progeny sired by mating *rnst-2*$^{\Delta/\Delta}$ males with WT females and microinjected with either anti-tDR or controls were evaluated for survival after heat-shock. $n = 3$ independent biological replicates, data are presented as mean ± SEM (unpaired two-tailed *t*-test ($P < 0.05$, ** $P < 0.01$). Source data are provided as a Source data file.

mating, with each biological replicate demonstrating increased survival (Fig. 5D), and overall statistical significance by Kaplan-Meier analysis (Log rank test $P = 9.83 \times 10^{-11}$–Supplementary Fig. 5G). This finding suggests that *rnst-2* males transmit potentially adaptive non-genetically inherited phenotypes to F1 progeny. Further experimentation on the adult F1 progeny of *rnst-2* males revealed that they are more sensitive to acute heat-shock than the progeny of *rnst-2* heterozygous males used as a control group (Fig. 5E). To determine changes to gene expression underlying this heat-shock sensitivity phenotype, we performed mRNA-seq on adult males and virgin females that were the progeny of *rnst-2* mutant and WT *fog-2* males crossed with WT *fog-2* females. Interestingly, amongst differentially expressed genes, we find several overlapping gene ontology (GO) terms that are dysregulated in both male and female progeny of *rnst-2* males, including germline, reproductive system, ribosome-associated terms, and nucleotide phosphate metabolic process (Supplementary Fig. 5C–E). We additionally analyzed heat-shock genes more specifically, assessing whether all genes with the gene name "*hsp*" displayed any coordinated regulation in *rnst-2* male mutant progeny. Consistent with our finding that *rnst-2* sired progeny are more sensitive to acute heat-shock, we find an overall decrease in the expression of *hsp* genes as a group in both male and female offspring compared to WT controls (Supplementary Fig. 5D, E). Using a similar analysis, we also found that ribosomal protein genes, annotated as gene names "*rpl*" and "*rps*," also exhibit coordinated decreased expression in *rnst-2* sired offspring, including both males and females, consistent with our GO analysis (Supplementary Fig. 5D, E). Overall, these findings indicate that *rnst-2* mutant males sire progeny with altered embryonic gene expression

that gives rise to inherited phenotypes during both larval development and adulthood.

## tDR Gly-GCC and Glu-CTC contribute to the transmission of phenotypes to offspring

While *rnst-2*$^{\Delta/\Delta}$ males and sperm produce progeny with altered embryonic gene expression and inherited phenotypes compared to control progeny, these experiments do not causally demonstrate that tDRs are responsible for the inheritance of the non-genetically inherited phenotypes. For example, other small RNAs regulated in *rnst-2* mutant sperm, such as rRNA-fragments or miRNAs (Fig. 3B), could additionally regulate post-fertilization gene expression and the transmission of phenotypes. To determine if specific tDRs are causally responsible for the transmission of phenotypes to progeny, we inhibited the function of tDR-Gly-GCC and tDR-Glu-CTC during fertilization by supplying anti-tDR (biotinylated RNAs antisense to 5'tRNA-halves) to the oocyte. This approach was previously and successfully employed in mouse preimplantation embryos to inhibit the functions of tDR-Gly-GCC in mice[5]. To perform this technique in *C. elegans*, we microinjected a cocktail of either anti-tDR-Gly-GCC, anti-tDR-Glu-CTC (20 ng/µL each), and GFP mRNA, or GFP mRNA only as a control, into the proximal germline of WT *fog-2* females that had been mated with *rnst-2*$^{\Delta/\Delta}$ males (Fig. 6A). We then screened females for GFP-positive embryos, which received the microinjection, and then either isolated these embryos for single-embryo mRNA-seq or allowed embryos to develop to adulthood for phenotyping. Supporting a direct function of tDRs transmitting non-genetically inherited phenotypes, in 2-cell embryos sired by *rnst-2* males with tDR function inhibited, we find a

rescue of *ife-2* upregulation (Fig. 6B and Supplementary Fig. 6A), as well as decreased expression of histones and *fbxb* genes in 8-cell embryos compared to control (GFP mRNA) injected *rnst-2* sired progeny (Fig. 6C, D). Similar to our comparison of *rnst-2* to WT male sired offspring, we find that the expression of all canonical histones exhibits coordinated rescue to WT RNA levels, specifically in 8-cell embryos, and not 2-cell embryos (Fig. 6C, D and Supplementary Fig. 6A, B). Importantly, we also find that inhibiting tDR function post-fertilization in *rnst-2* sired progeny rescues the sensitivity to heat-shock phenotype we found in *rnst-2* fathered sired offspring to resemble that of WT animals (Fig. 6E). These findings suggest that tDR-Gly-GCC and/or tDR-Glu-CTC in sperm causally regulate early embryonic gene expression and non-genetically inherited phenotypes in offspring.

## Discussion

Our findings reveal a previously unrecognized accumulation of specific tDRs in *C. elegans* sperm with functional relevance. In addition to regulating early embryonic gene expression, our data indicate that these tDRs contribute to the transmission of non-genetically inherited phenotypes to progeny, a mechanism that parallels observations in mammals. This functional conservation highlights the biological relevance of tDRs in sperm across diverse phyla of animals. We show that an RNaseT2 enzyme, *rnst-2*, regulates both the accumulation of specific tDRs as well as their length, as *rnst-2* mutants have increased levels of >30 nt tDRs compared to heterozygous or WT controls. Importantly, both a deletion (presumably null) mutant and a catalytic RNase mutant of *rnst-2* exhibit nearly identical molecular (small RNA/tDR alterations and changes in embryonic gene expression) and physiological (fertility) phenotypes, implying that catalytic activity of *rnst-2* is required for tDR regulation. We used these mutants to demonstrate that altered tDR accumulation in sperm leads to alterations in embryonic gene expression in sired progeny (Fig. 7). These post-fertilization functions of tDRs in regulating gene expression are highlighted by upregulation of the translational initiation factor *ife-2* at the 2-cell embryonic stage and suppressed histone and *fbxb* gene expression at the 8-cell embryonic stage. Altered embryonic gene expression in *rnst-2* sired progeny is additionally correlated with altered phenotypes, namely L1 starvation response and sensitivity to adult heat-shock. Importantly, we provide evidence that tDR Gly-GCC and Glu-CTC are causal for the inheritance of at least one of these phenotypes, as microinjection experiments inhibiting their functions block transmission of embryonic gene expression modulation and adult sequelae. Overall, these findings suggest that tDRs in *C. elegans* sperm transmit potentially adaptive inherited phenotypes from conception to adulthood, revealing that this mode of intergenerational non-genetic inheritance has broad conservation across members of the metazoan lineage, ranging from nematodes to mammals.

RNaseT2 enzymes have been previously implicated in tDR biogenesis in yeast under conditions of stress[30]. In this system, the RNaseT2 protein Rny1 was shown to be required for full-length tRNA cleavage into tRNAs-halves[30], which promotes cell death as well as interacts with ribosomes to regulate translation in response to stress[30]. Additionally, RNaseT2 enzymes in the ciliate *Tetrahymena* cleave tRNAs to generate tRNA halves under starvation conditions[56,57]. Further, in plants, RNaseT2 enzymes generate both tRNA-halves and shorter tDRs[58]. While these previous examples of RNaseT2-mediated tDR biogenesis rely on the direct cleavage of tRNAs to generate functional tDRs, our findings in *C. elegans* suggest that RNST-2 initiates the processing or degradation of tRNA-halves, generated by an unidentified RNase enzyme, into smaller (<30 nt) tDRs (Fig. 7). One of two mechanisms could explain these findings: (1) that RNST-2 enzymatically processes or degrades tRNA-halves into smaller RNAs or (2) that RNST-2 binds and stabilizes smaller fragments, meaning that in *rnst-2* mutants, smaller fragments are destabilized, and larger fragments (tRNA-halves) accumulate. While the binding and stabilization of smaller fragments cannot be ruled out, our extensive data demonstrates that a single catalytic amino acid mutant (*rnst-2^cat^(pen4)*) phenocopies null mutants, suggesting that the RNase activity of the enzyme is required for RNST-2 to regulate tDR accumulation.

Our data imply that a non-RNaseA (no orthologs in *C. elegans*) or non-RNaseT2 (only annotated ortholog is *rnst-2*) RNase enzyme acts as the initial *"cutter"* of tRNAs into tRNA-halves and that these tRNA-halves interact with RNST-2 to generate small tDRs or potentially eliminate them all together (Fig. 7). While we did not identify the initial *cutter* in this study, the identification of functional tDR accumulation and elucidation of a unappreciated step in biogenesis, trimming of tRNA-halves into small tDRs, establishes *C. elegans* as an attractive model system for determining factors involved in tDR biogenesis and regulatory functions. For example, beyond identifying the *cutter*, the effector proteins of tDRs have been elusive. In all characterized RNAi-

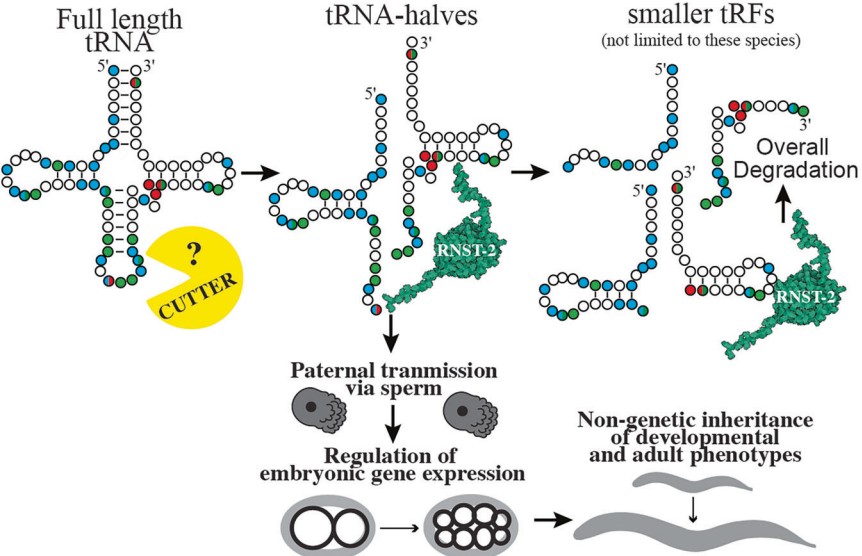

**Fig. 7 | Model for sperm tDR biogenesis and post-fertilization functions in *C. elegans*.** Schematic depicting how RNST-2 functions to regulate tDR accumulation and nucleotide length, as well as the post-fertilization embryonic and adult phenotypes transmitted by *rnst-2* modulated tDRs in *C. elegans*.

related pathways, small RNAs function in concert with Argonaute proteins as their effectors. While Argonautes have been postulated to bind tDRs in several models of tDR accumulation and function[19,59,60], binding to Argonautes from immunoprecipitation experiments is modest compared to canonical small RNAs bound (i.e., miRNAs) and, further, genetic evidence connecting them in coordinately regulating gene expression is lacking, beyond specific contexts[55,56]. An exception to this is 3' tDRs binding and functioning with Piwi-Argonautes in paramecium[24,25]. Other effector proteins of specific tDRs have been identified, such as the binding of effector protein HNRNP-F/H to 5'tDR-Gly-GCC in mESCs[26]. However, the ubiquity of this finding in other cell types and organisms has yet to be established. Importantly, it is unknown whether specific tDRs will bind explicit effector proteins or whether a common binding protein exists that uses the sequence specificity of different tDR sequences to confer specialized functions in gene regulation, such as in RNAi-related small RNA pathways. The vast uncertainty surrounding tDR biogenesis and function results from the study of these molecules being in its infancy (>20 years) and the absence of a robust genetic model to dissect their underlying molecular biology. To clarify the latter point, while much is known about the molecular mechanisms of tDR biogenesis and downstream gene regulation in yeast and mammalian cell-culture under conditions of stress, very little has been established as to how tDRs are generated and how they function in the germlines of animals. We predict that the use of *C. elegans* to study tDR biology in the germline, as well as in other aspects of organismal physiology, has the potential to transform our understanding of these small RNAs in the same way that the worm facilitated our understanding of RNAi and miRNAs.

Our finding that *rnst-2* regulates the processing of tRNA-halves into smaller RNAs, or their degradation altogether, reveals a previously underappreciated aspect of tDR biology: that the length of the tRNA-fragment matters. This suggests that the downstream gene regulatory functions of tDRs can be regulated at the level of tRNA cleavage into tRNA-halves, or by secondary RNA processing into smaller tDRs. Secondary processing has the potential to eliminate the functionality of longer RNAs, such as tRNA-halves or, alternatively, generate smaller RNAs with distinct functions. Our findings suggest that the functional regulatory tDRs in *C. elegans* sperm are tRNA-halves, as specifically inhibiting their function with antisense RNAs blocks their ability to regulate post-fertilization gene expression and transmit non-genetically inherited phenotypes. However, this result does not rule out the ability of tDRs processed by RNST-2 to have regulatory functions in other physiological contexts. In *C. elegans* sperm, our data support a model where RNST-2 degrades or processes specific tRNA-halves produced by the cleavage of mature tRNAs by a currently unknown ribonuclease into smaller tDRs or eliminates them altogether (Fig. 7). This processing would then function to reduce tRNA-halves and thus modulate their gene regulatory functions. We demonstrate here that the processing of tDRs by RNST-2 occurs in sperm, which, after fertilization, influences embryonic gene expression and the non-genetic inheritance of developmental and adult phenotypes. These findings introduce a novel concept that tDR length and the regulation of tDR processing influence their functions in the germline, revealing a tunable pathway capable of transmitting non-genetically inherited phenotypes from father to progeny.

Interestingly, in *C. elegans* females, *rnst-2* mutants do not exhibit elevated tDRs, possibly resulting from an absence of RNST-2 expression. Further, we find that *rnst-2* mRNA is targeted by abundant silencing endo-siRNAs (22G-RNAs), which we speculate silence *rnst-2* from being expressed in the female germline and oocytes and lead to the accumulation of tRNA-halves. These findings suggest that expression of *rnst-2* can be regulated to influence the processing of tRNA-halves, thereby altering overall tDR length and accumulation. Experiments in the future will assess whether environmental conditions, such as an altered diet, like in mammalian studies, can influence *rnst-2* in the

germline to regulate tDR processing and the transmission of non-genetically inherited phenotypes to progeny.

We demonstrate here that the regulation of tDR processing influences embryonic gene expression and affects the subsequent inheritance of developmental and adult phenotypes. Importantly, we show that tRNA-Gly-GCC and tRNA-GLU-CTC-derived 5' halves, regulated by RNST-2, modulate the mRNA expression of the translation initiation factor *ife-2* as well as histone and *fbxb* genes, at 2-cell and 8-cell stages of embryonic development, respectively. However, it is unclear exactly how these tDRs regulate the expression of these mRNAs, and further whether this regulation occurs by direct action of the small RNA on the mRNAs or indirectly. A variety of mechanisms in different systems have been suggested to underlie tDR regulatory functions, including transcriptional regulation, post-transcriptional gene regulation, and translational regulation. Interestingly, in mammals, it has been demonstrated that the 5'tDR-Gly-GCC regulates the abundance of histone RNAs through the regulation of U7 snRNA in cajal bodies[26]. While it is enticing to speculate that 5'tDR-Gly-GCC in *C. elegans* functions analogously, this is unlikely, as U7 snRNA has not been identified in worms, and the impact of 5'tDR-Gly-GCC on histone RNA regulation is opposite in the two organisms. 5'tDR-Gly-GCC leads to decreased expression of histone RNA in *C. elegans* and increased expression in mESCs and human embryonic stem cells. However, since sperm 5'tDR-Gly-GCC produce alteration in embryonic gene expression and offspring phenotypes in both mice and *C. elegans*, these findings raise the intriguing possibility that the modulation of histone RNAs during embryogenesis is a convergent mechanism underlying intergenerational epigenetic inheritance mediated by tDRs.

The other compelling regulatory mechanism previously demonstrated for tDRs is translation regulation reported in both mammalian and yeast cultured cells under conditions of stress, where tDRs can interact with the translation initiation and regulatory machinery to dampen translation globally[18,61]. Fascinatingly, in mammalian cell culture, 5'tRNA-halves derived from Alanine and Cysteine tRNAs were shown to physically interact with the eIF4F translational initiation factor to decrease translation initiation[55,61,62]. Our findings suggest that the ortholog of this gene, *ife-2*, is regulated at the RNA expression level by tDRs. While the direct molecular mechanisms of tDRs in these scenarios are clearly different, this could suggest that tDRs in both systems function to regulate translation. Additionally, direct translational regulation by sperm tDRs post-fertilization, which we do not capture in our data, could produce secondary effects such as altered histone and *fbxb* mRNAs and the inheritance of developmental and adult phenotypes. Unfortunately, ribosome profiling techniques to globally profile translation are currently not feasible in single (or low-input) *C. elegans* embryos. While these assays are potentially feasible in bulk embryo collections, they lack the resolution to determine the specific functions of tDRs in regulating translation in the very early stages of embryonic development. As techniques to profile translation progress, future studies will be performed to determine the effects of sperm tDRs on post-fertilization translation in *C. elegans*.

We demonstrate here that functional tDRs exist in *C. elegans* and specifically accumulate in sperm, analogous to tDR accumulation in sperm in mammals. Further, we discover a regulatory pathway where the RNaseT2 nuclease *rnst-2* modulates the accumulation and nucleotide length of tDRs, thereby promoting the clearance of specific tRNA-halves, including 5'Gly-GCC and 5'Glu-CTC. Enthrallingly, our data indicate that these tDRs, transmitted by sperm, regulate early embryonic gene expression and influence developmental and adult phenotypes in progeny. A similar phenomenon was previously observed in mammals, where sperm 5'Gly-GCC, regulated by paternal diet, modulates embryonic gene expression and facilitates the non-genetic inheritance of metabolic phenotypes in mice. These findings,

along with our own, suggest that the role of tDRs in intergenerational epigenetic inheritance is evolutionarily conserved. The emergence of *C. elegans* as a model to dissect the regulation of tDRs in the germline, their molecular functions in regulating early embryonic gene expression, and the subsequent inherited developmental alterations that produce inherited phenotypes has the potential to transform tRNA-derived RNA biology.

## Limitations of the study

While this study establishes a functional role for sperm tDRs in non-genetic inheritance, several technical and conceptual limitations should be considered when interpreting these findings. In particular, these limitations reflect the current lack of experimental strategies to selectively manipulate individual sperm RNAs and to biochemically define the enzymatic pathways that regulate their biogenesis. At present, antisense neutralization in the early embryo is the only experimentally tractable method to selectively disrupt individual sperm tDR species and assess their functional consequences, as no genetic or molecular strategy exists to specifically deplete individual tDRs in sperm prior to fertilization. This approach has been successfully employed in mammalian embryos to determine the functions of sperm delivered 5′Gly-GCC[5,26]. Complementary gain-of-function approaches, such as microinjection of synthetic tDRs, are unlikely to faithfully recapitulate endogenous tDR activity, as has been demonstrated in mammalian embryos, likely due to the requirement for native RNA modifications[4]. Indeed, the full repertoire and functional importance of tRNA and tDR modifications remain incompletely defined in *C. elegans*, precluding the generation of fully functional synthetic RNAs. Thus, while the antisense strategy used here provides strong and specific evidence for causality, future advances in tRNA modification mapping and manipulation will be required to expand the experimental toolkit for studying sperm tDR function. While the amplitude of the rescue we observe in our anti-tRF injection experiments is not completely WT, for example, noisy 2-cell expression of *ife-2* in anti-tRF injected embryos, our approach of adding anti-tDR to the maternal germline, and subsequently to the oocyte to inhibit sperm tDR function, is a technical challenge that, throughout *C. elegans* research, limits the determination of causality of RNAs transmitting phenotypes epigenetically across generations. Our approach here represents an important technical achievement and strongly supports the notion that tDRs causally regulate post-fertilization embryonic gene expression and the transmission of non-genetically inherited phenotypes.

A second limitation concerns the mechanistic role of RNST-2 in tDR regulation. Our genetic analysis using a catalytically inactive RNST-2 mutant strongly supports a direct role for RNST-2 enzymatic activity in shaping sperm tDR length and abundance. However, we were unable to biochemically demonstrate direct association of RNST-2 with mature tRNAs or tDR intermediates, as RNST-2 could not be reliably solubilized or purified for RNA immunoprecipitation assays despite extensive efforts. This technical limitation precludes definitive biochemical assignment of RNST-2 RNA substrates and opens the possibility that RNST-2 acts indirectly within a larger ribonucleoprotein or membrane-associated complex. Resolving this question will require the development of new biochemical approaches to access insoluble or compartmentalized RNaseT2 family members in vivo. Nevertheless, the convergence of genetic, small RNA profiling, and phenotypic data strongly supports a model in which RNST-2 plays a direct and essential role in regulating sperm tDR populations that transmit epigenetic information to the next generation.

## Methods

All research performed in this study complies with all relevant ethical regulations. This research was approved by the University of Pennsylvania Institutional Biosafety Committee (IBC#24-315).

### Materials availability statement

For additional information or requests for resources and reagents, please contact the lead investigator, Colin Conine (conine@upenn.edu), who will handle these inquiries. New strains generated in this study have been deposited at the Caenorhabditis Genetics Center.

### *C. elegans* culture and genetics

*C. elegans* culture and genetics were performed as previously described[63]. Unless otherwise noted, the "wild-type" (WT) strain used in this study was the Bristol N2 strain carrying the *fog-2(q71)* allele. Because *fog-2* hermaphrodites are incapable of producing sperm, we refer to *fog-2(q71)* hermaphrodites as "females" for clarity. Alleles used in this study are listed by chromosome. LGV: *fog-2(q71), rnst-2(pen1)*— deletion mutant, *rnst-2(pen3[gfp::3xflag::rnst-2]), rnst-2(pen4 [gfp::3xflag::rnst-2+c.178/179CA>GC])*—catalytic mutant. The *gfp::rnst-2* endogenous knock-in allele (*pen3*) was generated by streamlined genome engineering using self-excising drug selection cassette[64] Strains used in this study: CCC82 *rnst-2(pen1)/tmC16 [unc-60(tmIs1237)] V, fog-2(q71)*, CCC83 *pen4[GFP::3xflag::rnst-2+c.178/179CA>GC]/tmC16 [unc-60(tmIs1237)] V, fog-2(q71)*, CCC92 *pen3[GFP::3xflag::rnst-2] V, fog-2(q71)*.

### CRISPR/Cas9 genome editing

The *rnst-2* catalytic allele (*pen4*) was generated in the *gfp::rnst-2(pen3)* background as previously described[65]. In brief, sgRNA design was determined using GuideScan[66], which, in addition to Cas9 protein, trRNA, and single stranded 4nmol Ultramer oligos were ordered from IDT. The donor DNA template was 70 bp with 35 bp of homology on either side of the cut site and included the desired Histidine to Alanine substitution at amino acid 60. An injection mix was prepared consisting of 0.5 µL of Cas9 (10 µg/µL stock, 30 pmol), 5 µL of tracrRNA (0.4 µg/µL stock, 90 pmol), and 2.8 µL of crRNA (0.4 µg/µL stock, 95 pmol). The mixture was mixed and incubated at 37 °C for 15 min. 2.2 µL of ssODN donor (1 µg/µL stock) was then added followed by 1.6 µL of PRF4::rol-6 (su1006) plasmid (500 ng/µL solution) used as a co-injection marker. The final volume was brought to 20 µL with nuclease-free water, and the solution was spun at 13,000 × *g* for 2 min, and approximately 17 µL of the supernatant was reserved for germline injections of newly gravid *gfp::rnst-2* worms. After germline injection, approximately 30 F1 worms were singled from plates with the highest number of rollers, and Sanger sequencing was used to identify lesions of interest. This identical approach was used for indel generation (*pen1* allele) simply by omitting the addition of the ssDNA repair template.

### Brood size analysis

For *rnst-2* mutants, both deletion (*pen1*) and catalytic (*pen4*) mutants, heterozygous *rnst-2/tmc16;fog-2* males and females were mated to generate progeny whose parents had equivalent genotypes. Individual F1 L4 worms from these crosses, propagated at 20 °C, were then picked and mated with a single L4 animal of the reciprocal sex on a blank NGM plate with a small spot of OP50. Crosses were set up between *fog-2* X *fog-2*(WT), *rnst-2; fog-2* X *fog-2*, *fog-2* X *rnst-2; fog-2*, *rnst-2; fog-2/tmc16* X *fog-2*, and *gfp::rnst-2(pen3); fog-2* X *fog-2*. One day later, females full of eggs were transferred to NGM OP50 plates (plates with no eggs were considered unsuccessful matings and discarded). Each subsequent day, the animals were transferred to fresh plates until egg laying was complete. Plates were scored 1 day after removing the mother with eggs for the total number of progeny.

### Whole animal sample collection and RNA extraction

Unless otherwise stated, all RNA-seq experiments, beyond single-embryo experiments (below), were conducted on adult animals. To

generate *rnst-2* mutant males, heterozygous *rnst-2/tmc16* males and females were mated to generate progeny whose parents had equivalent genotypes. Individual (~50 total) F1 L4 homozygous and heterozygous animals from these crosses, propagated at 20 °C, were then picked as males and females separately to NGM OP50 plates. These animals were then developed at 20 °C for 24 h to generate adult animals and then picked into 1X M9 media with 0.5% poly-ethylene glycol (mPEG) in a 1.5 mL low adhesion microcentrifuge tube. Worms were washed three times with M9/mPEG and once with cold nuclease-free water before resuspending in 1 volume 2X worm lysis buffer (1% SDS, 80 mM Tris HCl, 400 mM NaCl, 20 mM EDTA, 0.8 mg/mL Proteinase K). The samples were then incubated at 55 °C for 15 min while shaking and visually inspected to ensure complete lysis.

Following worm lysis, 2X volume of TRI reagent (Invitrogen, AM9738) was added, followed by 1/5th volume of BCP (MRC, BP 151). This solution was mixed thoroughly and transferred to Phase Lock Gel tubes (QuantaBio, 2302830; pre-spun at 18,840 × *g* for 1 min) and spun at 18,840 × *g* for 4 min at 4 °C. The aqueous layer was then transferred to a new low-adhesion microcentrifuge tube, and 15 μg of GlycoBlue coprecipitant (Invitrogen, AM9515) was added prior to the addition of 1.1 volume of isopropanol. Samples were then precipitated at −20 °C for at least 1 h, and up to a month in isopropanol (longer storage times at −80 °C) to purify total RNA. To generate small RNA and mRNA-seq libraries, RNA samples in isopropanol were centrifuged at 18,840 × *g* for 15 min at 4 °C, then washed with cold 70% ethanol, spun again at 18,840 × *g* for 5 min at 4 °C, and finally reconstituted in nuclease-free water. RNA isolation from 50 adult males and females yields ~500 ng and ~1 μg, respectively.

### Northern blot
Northern blotting was performed as described previously[48]. Briefly, 5 μg of total male RNA per sample was resolved on a denaturing 15% polyacrylamide–urea gel and transferred to a positively charged nylon membrane. RNAs were UV crosslinked and hybridized overnight at 68 °C with digoxigenin-labeled DNA oligonucleotide probes (sequences in Supplementary Information) in ULTRAhyb-Oligo buffer. Membranes were washed under high-stringency conditions and probed with anti-digoxigenin alkaline phosphatase–conjugated antibody (1:1000 dilution anti-digoxigenin-AP, Fab fragments - Roche, Cat #110932774910). Signal was detected using chemiluminescent substrate and imaged on a Bio-Rad ChemiDoc MP system. Band intensities were quantified using ImageJ. Membranes were stripped and re-probed with U6 as a loading control.

### Small RNA-seq library preparation
Two hundred nanograms of total RNA were used as input for all small RNA sequencing experiments, with library construction carried out using a modified Illumina Tru-Seq small RNA library prep protocol (Illumina RS-200). Small RNAs (18–40 nt) were first purified by size selection from denaturing 15% polyacrylamide 7 M urea TBE gel electrophoresis. Small RNAs were eluted in 10 mM Tris-HCl, pH 7.5, 300 mM NaCl, and 1 mM EDTA, shaking overnight at room temperature. Small RNAs were then precipitated by separating the gel debris using 0.45 μM cellulose-acetate filters, adding 1 μL GlycoBlue and 1.1 volume of isopropanol, mixing well, and placing the samples at −20 °C for at least 1 h. The samples were then centrifuged at 18,840 × *g* for 15 min at 4 °C, and the supernatant was discarded before washing with 1 mL of cold 70% ethanol. Following an 18,840 × *g* spin at 4 °C for 5 min and after complete ethanol removal, the pellet was reconstituted in RNA 5′ Pyrophosphohydrolase (RppH - NEB M0356S) per manufacturers guidelines and incubated at 37 °C for 1 h. Following RppH treatment, small RNAs were precipitated and ligated to the 3′ adapter using Truncated T4 RNA ligase (Lucigen LR2D11310K). Small RNAs

were then ligated to the 5′ adapter using T4 ligase, and subsequently reverse transcribed using Superscript II (Invitrogen 18064014) following the Tru-Seq small RNA protocol. Following PCR optimization, to determine the proper number of cycles for amplification, the libraries were amplified, gel purified on an 8% TBE polyacrylamide gel (to remove cloned products with no insert) and combined at equimolar concentration. Libraries were then sequenced using an Illumina NextSeq1000 (75-bp single read).

### mRNA-seq library preparation
mRNA-seq library construction was carried out using the Illumina stranded mRNA kit as per the manufacturer's instructions (Illumina 20040534). In brief, poly-adenylated RNAs were captured using oligo(dT) magnetic beads from 200-800 ng total RNA. The RNA was then fragmented, primed for cDNA synthesis, and subsequently cleaned up using AMPure XP beads (Beckman Coulter, Cat# A63881). Following PCR optimization, to determine the proper number of cycles for amplification, the libraries were amplified and combined at equimolar concentrations before being purified using non-denaturing PAGE. Paired-end sequencing (50 bp per read) was performed using an Illumina NextSeq 1000.

### Embryo isolation and single-embryo mRNA-sequencing
Cross plates were prepared by placing 50 WT (*fog-2(q71)*) L4-stage females and 50 *rnst*-2^Δ/Δ (*pen1*); *fog-2(q71)* L4-stage males on 60 mM NGM plates seeded with OP50. The next day, mated females full of eggs were picked and washed several times with M9 mPEG and then subjected to a 5-min hypochlorite treatment. Early-stage embryos, identified under a stereomicroscope, were then transferred to 10 μg/μL chitinase in M9 mPEG treatment and incubated at room temperature for 8–10 min. Following chitinase treatment, 2-cell and 8-cell embryos, identified visually under a dissecting stereomicroscope, were placed in RNA worm lysis buffer in PCR tubes and incubated at 55 °C for 10 min. Embryo lysates were then stored at −80 °C before proceeding to library preparation. Single-embryo mRNA-seq libraries were constructed using a SMART-Seq protocol[67,68] used previously for single mouse preimplantation embryos[69,70]. RNA was isolated with RNAClean-XP beads (Beckman Coulter, Cat# A63987), and full-length polyadenylated RNA was reverse transcribed using Superscript II (Invitrogen, Cat# 18064014). The cDNA was then amplified for 10 cycles and subsequently 0.33 ng of cDNA was used to construct a pool of uniquely indexed samples with the Nextera XT kit (Illumina, Cat# FC-131-1096). Finally, pooled libraries were sequenced on a NextSeq1000 with paired-end sequencing (50 bp per read).

*rnst-2* mutant and WT type matings were set up with 4 independent groups of worms on separate days for embryo collection (4 biological replicates). For each embryo collection day, 5–8 embryos per group (mutant or WT father) and embryonic stage were isolated, for a total of 20–30 embryos sequenced per condition.

### *C. elegans* sperm isolation
For preparation of sperm samples used in small RNA sequencing, *fog-2 C. elegans* were grown on 150 mm NGM plates supplemented with 10% agarose. Worms were synchronized as L1 larvae and resuspended in concentrated OP50 at 100,000 worms per mL. 1 mL of this OP50 solution was plated on 150 mm plates and allowed to dry before being placed in a 20 °C incubator. Approximately 3 days later, worms were washed off 5–8 plates, and a series of gravity floats in M9/mPEG were performed to rid the sample of bacteria and any L1 larvae. Males were then filtered using a 35 μm mesh above a pool of M9/mPEG for approximately 15 min checking the sample under the mesh for male purity. Starting with a worm population of >95% males and at least 100,000 males (for *rnst-2* mutants, the worm population was at least

>90% because synchronization was harder to fully maintain), the worms were concentrated into a volume of approximately 500 μL and pressed between two plexiglass plates for ~10 min. M9/mPEG was then used to wash liberated sperm and crushed males into a 50 mL conical tube. This solution was then filtered using a 10 μm mesh and spun at $800 \times g$ for 10 min to pellet sperm. To remove cellular debris, the sperm were washed 1–2 additional times by spinning at $600 \times g$ for 5 min. Following a microscopic evaluation of purity, the sperm were transferred to a 1.5 mL Eppendorf tube, pelleted by centrifugation, and washed with cold water. The sample was brought to 50 μL, and an equal volume of 2X worm lysis buffer with DTT was added. Sperm were lysed, and RNA was extracted as described for whole worms.

## Heat shock survival assay

Crosses were set up between L4-staged male and female worms for heterozygous (*rnst-2$^{cat/+}$(pen4);fog-2* X *fog-2*) and *rnst-2$^{cat/cat}$(pen4); fog-2* X WT *fog-2* to generate F1 progeny. One 60 mM NGM plate seeded with OP50 was set up for each condition, with 50 L4 females and 50 males per plate. One day later, females full of eggs were transferred to new OP50 NGM plates (10 per plate) and allowed to lay eggs for 4 h at 20 °C. Subsequently, all adult females were removed, and progeny proceeded to develop at 20 °C for 50 h. Plates were then wrapped in parafilm and submerged in a pre-heated 37 °C water bath for 85 min. Following heat shock plates were allowed to recover at 20 °C for 24 h and then assessed for survival by touching the animals with a worm pick. For each experiment, 3 technical replicates (plates of progeny) were counted with at least 50 worms per plate. These experiments were further biologically replicated 3 times, on separate days of growing worms and setting up crosses.

## L1 starvation survival

Crosses were set up between L4-staged male and female worms for WT (*fog-2* X *fog-2*) and *rnst-2$^{cat/cat}$ (pen4); fog-2* X WT *fog-2* to generate F1 progeny. A total of three 60 mM NGM plates seeded with OP50 were set up for each condition, with 50 L4 females and 50 males per plate. After 2 days, worms were subjected to hypochlorite treatment to isolate pure populations of embryos. Embryos in M9/mPEG at a density of 10–20 embryos per μL were placed on a rocker at room temperature. Worms were then counted daily and scored for percent survival (dead/living L1s) by taking 10–20 μL of M9/mPEG with L1 progeny and plating them on an NGM OP50 plate. Each day, 3 plates per condition were counted as technical replicates. These experiments were further biologically replicated 5 times from separate days of growing worms and setting up crosses.

## *C. elegans* germline microinjections with anti-tDR oligos

*rnst-2$^{Δ/Δ}$(pen1); fog-2* males were mated to WT *fog-2* females, by placing 50 L4 worm of each sex on 60 mM NGM OP50 seeded plates (typically 3–4 plates). Approximately 16 h later, the germline of females with embryos was microinjected with RNA[71]. RNA microinjected included antisense-tDR-5′Gly-GCC and antisense-tDR-5′Glu-CTC, both RNAs representing the reverse complement of the 5′tRNA halves synthesized by IDT, as well as *his-72::gfp* RNA for anti-tDR injections, and *his-72::gfp* RNA only for control injections. RNA injections were carried out using a Femtojet (Eppendorf) microinjector at 2000 hPa until visible expansion of the germline was observed. Anti-tDRs and *his-72::gfp* RNA were injected at a concentration of 20 ng/μL each and 30–50 ng/μL, respectively. *his-72::gfp* RNA was made by cloning 1 kb of the upstream and downstream of zuIs178 from the strain RW10029[72] into pCR-Blunt II-TOPO (ThermoFisher, Cat# K280002) and generating RNA for injection via in vitro transcription (IVT). Six to twelve hours post-injections, RNA-injected animals were assessed for GFP expression in embryos under a fluorescent stereomicroscope. These animals were then hypochlorite-treated to isolate embryos, and subsequently GFP-positive embryos were chitinase-treated for single-embryo RNA-seq (2- & 8-cell embryos–above) or transferred to NGM OP50 plates and allowed to develop to adulthood for heat shock survival experiments (described above).

*rnst-2$^{Δ/Δ}$(pen1); fog-2* males X *fog-2* females crosses and micro-injection were performed on 4 independent groups of worms on separate days for embryos collection and heat shock survival (4 biological replicates). For single-embryo RNA-seq, 4–8 embryos per group (anti-tDR or control injected) and embryonic stage were isolated per collection day, for a total of 18–26 embryos sequenced per condition. For heat shock survival, 2 technical replicates of 20–50 worms were assayed per condition and biological replicate.

anti-tDR-5′Gly-GCC /5BiosG/rArGrGrCrGrArGrCrArUrUrCrUr ArCrCrArCrUrGrArArCrCrArCrCrGrArUrGmC

anti-tDR-5′Glu-CTC /5BiosG/rArGrGrCrCrArUrArArArUrCrCrUrArArCrCrArCrUrArGrAr CrCrArCrArArCrGrGmA

## Bioinformatic analysis of mRNA-seq and small RNA-seq libraries

Bioinformatic pipelines were assembled using the Via Foundry, formerly DolphinNext, pipeline generator for all RNA-sequencing data analyzed in this study[73]. mRNA libraries were first aligned to the *C. elegans* genome (WS245) using STAR and quantified and normalized to transcripts per million (TPM) using RSEM. Differential gene expression was calculated with fold-change of expression and *t*-tests (unpaired, two-tailed), identifying up- and downregulated genes using the following cutoff: $\log_2$(fold change) >1 or less than −1 and a *P* value ≤ 0.01 (unless otherwise specified). For differentially expressed genes, WormBase enrichment analysis was used to determine tissue, pathway, and GO enrichment[74,75].

Small RNA libraries were trimmed of adapter sequences using trimmomatic and aligned sequentially (in order) to annotated *C. elegans* rRNAs, miRNAs, tRNAs, and piRNAs using Bowtie 2. STAR was used to align small RNAs to the *C. elegans* genome (WS245), and featureCounts was then used to quantify reads mapping to genomic features (mRNA fragments and endo-siRNA). Mapped reads were normalized to reads per million using custom Python scripts, to total genome mapping reads, genome mapping reads–rRNA mapping reads, or to a specific RNA class, for subsequent analysis. Where indicated in the text, 22 G RNAs targeting the *rnst-2* gene locus were extracted from small RNA-seq data and used to visualize alignments. These reads were parsed from BAM files using a custom Python script and defined as antisense reads 22 nt long, with G as the 5′ nucleotide.

## Reporting summary

Further information on research design is available in the Nature Portfolio Reporting Summary linked to this article.

# Data availability

Sequencing data generated in this study are available in the Gene Expression Omnibus (GEO) under accession code GSE293020. RNA sequencing data for the figures are provided in the Supplementary Tables. The source data underlying the figures are provided in the Source data file. Source data are provided with this paper.

# Code availability

RNA-seq and bioinformatics analyses were performed using the DolphinNext/Foundry pipeline builder[73]. All graphing and statistical analyses were performed using GraphPad Prism and Python. All code used for data processing and analysis is publicly available at GitHub (https://github.com/conine-lab/trna-derived-rna-celegans-analysis) and archived at Zenodo (https://doi.org/10.5281/zenodo.18468451).

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

## Acknowledgements

The authors thank Conine lab members and Taku Kambayashi for critical reading of the manuscript, as well as the Penn Worm Group for feedback on interpretation and presentation of data. This work was supported by a Pew Biomedical Scholars Award to C.C.C. and a NIH/NIGMS MIRA/R35 (R35GM151087) awarded to C.C.C. B.K.S. was supported by a PennPort fellowship through NIH/NIGMS (K12GM081259).

## Author contributions

Conceptualization: N.S.G., O.J.C., and C.C.C.; methodology: N.S.G., O.J.C., A.E.S., O.Y., A.K., and C.C.C.; investigation: N.S.G., O.J.C., B.K.S., K.S.A., K.E.G., A.E.S., J.L., and C.C.C.; formal analysis: N.S.G., O.J.C., O.Y., A.K., and C.C.C.; writing—original draft: N.S.G., O.J.C., and C.C.C.; writing—review and editing: N.S.G., O.J.C., A.E.S., B.K.S., A.K., and C.C.C.; supervision: A.K. and C.C.C.; funding acquisition: B.K.S., A.K., and C.C.C.

## Competing interests

A.K. is a co-founder of Via Scientific, Inc., a UMass Chan Medical School spin-off. A.K. is a board member of the company. A.K. and O.Y. have equity in the company. They ensure that steps have been taken to prevent these affiliations from affecting analysis integrity and are dedicated to upholding research transparency and integrity. The remaining authors declare no competing interests.
