## [Transparent Peer Review file · Nature Communications]

tRNA-derived RNA processing in sperm transmits non-genetically inherited phenotypes to offspring in *C. elegans*

Corresponding Author: Dr Colin Conine

Version 0:

Reviewer comments:

Reviewer #1

(Remarks to the Author)

tRNA-derived RNAs (tDRs) are thought to be important mediators of epigenetic inheritance in mammals. At the same time, while epigenetic inheritance is well established phenomenologically and mechanistically in *C. elegans*, tDRs have not been shown to play a role in the *C. elegans* system. In this report, Galambos et al establish the presence of tDRs in sperm of *C. elegans*, identify the RNase T2 *rnst-2* as a regulator of this class of small RNAs, characterize molecular features of this class of RNAs, and provide evidence suggestive of a role in intergenerational influences on gene regulation and organismal responses to stress. This is an important advance in the field of epigenetic inheritance as it establishes *C. elegans* as a model system for studying tDR contributions and indicates deep conservation of this mechanism in metazoa.

Together, the evidence presented in this study supports the conclusions that tDRs are present in *C. elegans* sperm, that they are regulated by *rnst-2*, and that they affect early embryonic gene expression. However, the claims about intergenerational effects on phenotype, and related inferences about overall biological significance, are substantially weaker. In particular, the rescue experiments with tDR-Gly-GCC and tDR-Glu-CTC anti-tDRs are not very convincing. Effects of the mutants on tDR abundance are often relatively small, and the loss of *rnst-2* appears to affect miRNAs (or at least specific miRNAs), which is likely to affect mutant phenotypes, but this is not discussed. In addition, rigor should be improved throughout the paper by adding significance tests for the effects reported, especially because these effects are sometimes relatively small in magnitude. Overall, this is an important paper but as written, does not fully support all claims with the evidence presented.

Major points:

- 1) In Fig. 1A, the plot makes it appear that mRNAs are also enriched in sperm. Alternatively, this could be because most of the black points are hidden behind other data points. This effect should be explained or the data should be more clearly presented.
- 2) In Fig. 1E, the relative accumulation of 5' vs. 3' fragments in sperm seems clear for Ser-AGA and Glu-CTC, but not as much for Gly-GCC as claimed in the text. This claim (and all claims in this figure) should be supported statistically.
- 3) It would be useful to know what fraction of individual tDRs in sperm have a 5' or 3' fragment bias. This is a somewhat different question from the analysis in Fig. 1D, which is the overall bias across the whole class.
- 4) For generation of the *rnst-2* deletion allele in Fig. 2/S2, some data confirming that the gene product is depleted, at least at the transcript level (lack of full-length coding transcript) or even better at the protein level, should be provided.
- 5) How similar are the sets of upregulated tDRs in the *rnst-2* deletion and catalytic mutant backgrounds? Data is shown for classes of small RNAs for each mutant separately, but not the specific tDRs.
- 6) In Fig. 3, the size of the effect on tDR enrichment in mutant sperm appears relatively small. From Fig. 3B, it looks like there is also an increase in miRNAs in the mutant, but this does not appear to be reflected in Fig. 3D. How can this be reconciled? If miRNAs also increase in the mutant, this is likely to affect the phenotype.
- 7) In many figures, claims are made about a change in a specific RNA population without statistics (for example, proportions

of different RNA classes in Fig. 1B, C, and D, and all similarly formatted figures throughout the manuscript). Statistical tests should be done to support these claims. This is especially important when comparing the analysis in females (Fig. 4) to males, since it's hard to assess by eye whether the small changes observed in females are less significant than the somewhat larger changes observed in males.

8) In Fig. 5B, *ife-2* is highlighted as an upregulated gene in 2C embryos derived from *rnst-2* mutant fathers, but it is not clear why exactly this gene is selected since it is not the most upregulated. Relatedly, the effect of anti-tDR injection on *ife-2* transcript in Fig. 6B is small and highly variable, raising doubts about the validity of this gene as a regulatory target of sperm tDRs.

9) The effect on L1 starvation shown in Fig. 5D is small and does not strongly support the conclusion that "*rnst-2* males transmit potentially adaptive non-genetically inherited phenotypes to F1 progeny". While the effect on heat shock response seems larger, the associated decrease in hsp gene expression is also quite modest. These results are suggestive, but do not support a strong conclusion about intergenerational effects on phenotype.

10) There are several problems with the anti-tDR injection experiment shown in Fig. 6. Fig. 6B shows a small and highly variable effect on *ife-2* expression, and Fig. 6C does not convincingly show rescue of the histone and *fbxb* genes compared to Fig. 5C. In addition, no data is shown to indicate how much anti-tDR was injected, or what effect this had on actual target tDR levels. The strong conclusions about the specific roles of tDR-Gly-GCC and tDR-Glu-CTC in the Results and Discussion sections, including: "These findings demonstrate that tDR-Gly-GCC and/or tDR-Glu-CTC in sperm causally regulate early embryonic gene expression and non-genetically inherited phenotypes in offspring" and "we demonstrate that tDR-Gly-GCC and tDR-Glu-CTC are causal for the inheritance of these phenotypes" are not warranted based on the data shown. Overall, the study would be stronger with this set of experiments removed.

Minor:

11) The assertion that tDR function in sperm is "deeply conserved among metazoans" at the end of the introduction is a stretch based on only two species.

12) In the first paragraph of the results, it would be useful to specify that previous studies were performed on whole adult hermaphrodite worms (vs. "adult hermaphrodite worms"), for readers who are accustomed to working with isolated adult mammalian tissues.

13) This pair of sentences in the first paragraph of the Results section is very confusing: "The *fog-2* mutation produces hermaphrodites that do not undergo spermatogenesis yet produce normal oocytes. Further, *fog-2* is not expressed or required for any aspect of male gametogenesis, thus producing normal sperm". Initially this appears contradictory, but on re-reading I think it means that hermaphrodites produce only oocytes, while males produce (only) normal sperm. If correct, this could be explained more clearly.

14) In Fig. 3A, there are non-sperm GFP+ cells in the bottom image, which appears to contradict the claim in the text that GFP::*RNST-2* is expressed specifically in sperm. I suspect that the source of the additional GFP is autofluorescence or a coinjection marker. If so, this should be specified in the legend. If it is from real transgene expression outside the reproductive tract, this should be more clearly explained in the text and the conclusion should be adjusted.

15) Please add page and line numbers to make comments easier to reference.

Reviewer #2

(Remarks to the Author)

It is known that in mouse models, the diet of males regulates production of specific tRNA-derived RNAs (tDRs) in sperm. While some mechanistic insights were obtained on the action of these tDRs in rodents, the details of their biogenesis are still missing. In this work, Galambos et al used worm model to get insights into similar phenomenon and determined the RNase T2 enzyme, *rnst-2*, as a regulator of *C. elegans* tDR accumulation. RNY1, yeast RNase T2, is shown to cleave tRNAs in the anticodon to produce tDRs. In worms, T2 does not cut the tRNA initially but regulates production of shorter tDRs, likely from tRNA halves.

The main question still remains- what is identity of the tRNase that makes first cut. Nonetheless, the data provided in this work is appealing.

My major concerns are that all of this work is based on RNA-seq. I would like to see a northern blotting and qRT-PCR to back up the data. The same is true for the experiments that are based on anti-tDR oligos, namely what is the efficiency of the depletion, what happens with mature tRNAs, are their levels affected?

The experiment with catalitically dead *rnst-2* is very appealing. Authors should use this mutant for "CLIP/RIP-like" experiments to demonstrate whether the enzyme directly pulls mature tRNAs or their cleavage variants, I believe that this would be very important to figure out.

Finally, the effect of tDR depletion on gene expression, especially on worm eIF4E are interesting. Would injection of synthetic tDRS and their mutants affect ife-2, histone and fbx genes?

Version 1:

Reviewer comments:

Reviewer #1

(Remarks to the Author)

The authors have addressed my concerns and I am happy with the improved manuscript.

I have one remaining minor adjustment to request. Since rescue data is only shown for one of the two phenotypes tested in progeny (Fig. 6E shows rescue of heat shock sensitivity, but L1 survival is not tested), the text should better acknowledge this. Specifically, line 449-450 in the Discussion (“we provide evidence that tDR Gly-GCC and Glu-CTC are causal for the inheritance of these phenotypes”) would be better phrased as “we provide evidence that tDR Gly-GCC and Glu-CTC are causal for the inheritance of at least one of these phenotypes”.

Reviewer #2

(Remarks to the Author)

I understand the technical issues that authors faced with. All other issues were addressed adequately. I recommend the work for publication

Response to reviewers for manuscript entitled: “tRNA-derived RNA processing in sperm transmits non-genetically inherited phenotypes to offspring in *C. elegans*”.

On behalf of the authors, I wish to thank the reviewers for their expert review of our manuscript and the opportunity to address their comments. We have undertaken a comprehensive revision of the manuscript, having now addressed all the reviewer’s comments, we believe the manuscript is very much improved through their insight and suggestions. We have included a letter that responds to each point raised by the reviewers. In addition, we have uploaded both a clean copy of the revised manuscript and a marked-up copy in which our changes have been highlighted with track changes. Please note that the page/line numbers listed below in our Response to Reviewers reflect those in the marked-up version of the manuscript.

Kindest regards,
Colin Conine, Ph.D.

Reviewer #1: tRNA-derived RNAs (tDRs) are thought to be important mediators of epigenetic inheritance in mammals. At the same time, while epigenetic inheritance is well established phenomenologically and mechanistically in *C. elegans*, tDRs have not been shown to play a role in the *C. elegans* system. In this report, Galambos et al establish the presence of tDRs in sperm of *C. elegans*, identify the RNase T2 *rnst-2* as a regulator of this class of small RNAs, characterize molecular features of this class of RNAs, and provide evidence suggestive of a role in intergenerational influences on gene regulation and organismal responses to stress. This is an important advance in the field of epigenetic inheritance as it establishes *C. elegans* as a model system for studying tDR contributions and indicates deep conservation of this mechanism in metazoa.

Together, the evidence presented in this study supports the conclusions that tDRs are present in *C. elegans* sperm, that they are regulated by *rnst-2*, and that they affect early embryonic gene expression. However, the claims about intergenerational effects on phenotype, and related inferences about overall biological significance, are substantially weaker. In particular, the rescue experiments with tDR-Gly-GCC and tDR-Glu-CTC anti-tDRs are not very convincing. Effects of the mutants on tDR abundance are often relatively small, and the loss of *rnst-2* appears to affect miRNAs (or at least specific miRNAs), which is likely to affect mutant phenotypes, but this is not discussed. In addition, rigor should be improved throughout the paper by adding significance tests for the effects reported, especially because these effects are sometimes relatively small in magnitude. Overall, this is an important paper but as written, does not fully support all claims with the evidence presented.

We thank the reviewer for their acknowledgement of our important advance in the field and their assessment and suggestions for the manuscript. We hope that the reviewer

finds our responses to their comments and our amendments to the manuscript satisfactory.

Major points:

1) In Fig. 1A, the plot makes it appear that mRNAs are also enriched in sperm. Alternatively, this could be because most of the black points are hidden behind other data points. This effect should be explained or the data should be more clearly presented.

We thank the reviewer for raising this point, indeed mRNA-fragment (denoted as fragments because they are cloned during a small RNA-seq protocol) accumulate in sperm. This is a common phenomenon in animals that occurs as sperm differentiate at the completion of spermatogenesis. We now acknowledge these mRNA-fragments and discuss this in the manuscript on Page 4, lines 173-176.

*As previously reported we additionally we find that mRNA-fragments are also enriched in *C. elegans* sperm, which are likely the remnants of spermatogenic mRNA degradation which occurs as sperm undergo final steps of maturation in most animals^{45,46}.*

2) In Fig. 1E, the relative accumulation of 5' vs. 3' fragments in sperm seems clear for Ser-AGA and Glu-CTC, but not as much for Gly-GCC as claimed in the text. This claim (and all claims in this figure) should be supported statistically.

We have now provided statistics, where appropriate, to support all claims made in this figure (also see Major Comment #7 below).

3) It would be useful to know what fraction of individual tDRs in sperm have a 5' or 3' fragment bias. This is a somewhat different question from the analysis in Fig. 1D, which is the overall bias across the whole class.

*We thank the reviewer for this great idea. We performed this analysis for all tDR mapping reads from our sperm small RNA-seq data, which reveals an interesting distribution of 5' and 3' tDRs derived from different tRNAs (**Supplementary Figure 1E**). Additionally, we now discuss this in the manuscript on Page 5, lines 211-213.*

*Interestingly, individual tRNA isoacceptors display a wide range of 5' and 3' fragment biases, suggesting isoacceptor-specific differences in tDR biogenesis and stability in sperm (**Supplementary Figure 1E**).*

4) For generation of the *rnst-2* deletion allele in Fig. 2/S2, some data confirming that the gene product is depleted, at least at the transcript level (lack of full-length coding transcript) or even better at the protein level, should be provided.

*This is an important point raised by the reviewer that should certainly be addressed. We have now performed mRNA-seq on $rnst-2^{\Delta/\Delta}$ and $rnst-2^{cat/cat}$ males, as well as heterozygous controls (**Supplementary Figure 2B**). We discuss this in the manuscript on Page 5, lines 237-239.*

*Importantly, homozygous deletion mutants show a nearly complete loss of $rnst-2$ expression relative to heterozygous males, while the catalytic point mutations retain partial expression of the gene (**Supplementary Figure 2B**).*

5) How similar are the sets of upregulated tDRs in the $rnst-2$ deletion and catalytic mutant backgrounds? Data is shown for classes of small RNAs for each mutant separately, but not the specific tDRs.

*This is another great point raised by the reviewer and an analysis we often do in our lab. We have now made a comparison plot where we graph the fold-change of all small RNA values (rpm normalized to total genome mapping reads) for $rnst-2^{cat/cat}$ versus $rnst-2^{cat/+}$ males on the x-axis and $rnst-2^{\Delta/\Delta}$ versus $rnst-2^{\Delta/+}$ on the y-axis (**Supplementary Figure 2D**). Interestingly, we find a striking correlation for tDR regulation on the mutants ($r = 0.73$, $P < 0.0001$), rRNA-fragments ($r = 0.98$, $P < 0.022$), and mRNA-fragments ($r = 0.69$, $P < 0.0001$). This is now discussed in the text on Page 6, lines 243-246.*

*Further we find a striking correlation in the accumulation of tDRs, rRNA-fragments, and mRNAs-fragments in both $rnst-2^{\Delta/\Delta}$ and $rnst-2^{cat/cat}$ males compared to heterozygous controls, suggesting similar functional consequences on these RNA species in distinct $rnst-2$ mutations (**Supplementary Figure 2D**).*

6) In Fig. 3, the size of the effect on tDR enrichment in mutant sperm appears relatively small. From Fig. 3B, it looks like there is also an increase in miRNAs in the mutant, but this does not appear to be reflected in Fig. 3D. How can this be reconciled? If miRNAs also increase in the mutant, this is likely to affect the phenotype.

*The reviewer raises a valid point. While some miRNAs increase in mutant sperm and others decrease, many of the upregulated miRNAs are lowly abundant, whereas the downregulated miRNAs (although fewer) are substantially more abundant, which accounts for the overall consistency observed in **Figure 3D**. miRNAs in sperm could potentially regulate post-fertilization gene expression and phenotypes in progeny, which we now address in the manuscript on Page 9, lines 423-425.*

*For example, other small RNAs regulated in $rnst-2$ mutant sperm, such as rRNA-fragments or miRNAs (**Figure 3B**), could additionally regulate post-fertilization gene expression and the transmission of phenotypes.*

7) In many figures, claims are made about a change in a specific RNA population without statistics (for example, proportions of different RNA classes in Fig. 1B, C, and D, and all similarly formatted figures throughout the manuscript). Statistical tests should be done to support these claims. This is especially important when comparing the analysis in females (Fig. 4) to males, since it's hard to assess by eye whether the small changes observed in females are less significant than the somewhat larger changes observed in males.

We thank the reviewer for this point because statistics were certainly something lacking in our original manuscript. We have now added statistical analysis to support all claims we make throughout the manuscript in the figures, figure legends, and at times (when relevant) in the text. Regarding specific points raised by the reviewer we have performed replicate-level statistical analyses for all figures showing small RNA class proportions, 5'3'tDR distributions, RNA length and first nucleotide distributions, and tDR accumulation across mature tRNAs, including Fig. 1B–D and all similarly formatted figures throughout the manuscript. For each biological replicate, small RNA class proportions were calculated independently based on aligned sequencing reads, and statistical comparisons between conditions were performed using two-sided Mann–Whitney U tests. Where multiple comparisons were performed (e.g., across RNA classes or length bins), Benjamini–Hochberg FDR correction was applied. Importantly, the same statistical framework was applied consistently to both male and female datasets (including Fig. 4), allowing direct assessment of whether observed differences in females are statistically weaker than those observed in males. Figure legends and the Methods sections have been updated accordingly to explicitly state the statistical tests used.

8) In Fig. 5B, *ife-2* is highlighted as an upregulated gene in 2C embryos derived from *rst-2* mutant fathers, but it is not clear why exactly this gene is selected since it is not the most upregulated. Relatedly, the effect of anti-tDR injection on *ife-2* transcript in Fig. 6B is small and highly variable, raising doubts about the validity of this gene as a regulatory target of sperm tDRs.

**ife-2* was focused on because it was the most statistically significant gene regulated ($P > 0.000001$) in 2-cell *rst-2* sired embryos. It was also interesting to us because it is a translational regulator that has been shown to interact with tDRs in the literature. We have now clarified this on Page 8, lines 368-370.*

Among these, *ife-2*, which encodes an eIF4E translation initiation factor, was the most significantly affected gene and was upregulated 2.5-fold ($P < 0.5 \times 10^{-5}$ – **Figure 5B & Supplementary Figure 5A).**

*Regarding the variable regulation of *ife-2* in our anti-tDR injection experiments, while the regulation is significant ($P = 0.0003$), we agree that embryos respond variably. We hypothesize that this variability arises from the nature of the experimental technique. We injected anti-tDRs into the female germline, where they are packaged into oocytes and subsequently bind sperm-delivered tDRs to block their function. We track delivery of anti-tDRs to oocytes and embryos by co-injecting GFP mRNA, and while this allows us to assess which embryos received an injection, it does not provide sufficient resolution in GFP expression to determine the delivered dose. Thus, embryos in these experiments likely receive variable exposure to anti-tDRs. While this is certainly not a perfect experiment, it is currently the only feasible approach (and, to our knowledge, the first time such an experiment has been performed) to assess direct causality for sperm tDRs in post-fertilization functions and in the transmission of non-genetically inherited phenotypes. Thus, we believe this approach represents an important first step toward experimentally probing tDR function in early embryos and provides a foundation for future methodological improvements.*

9) The effect on L1 starvation shown in Fig. 5D is small and does not strongly support the conclusion that “*rnst-2* males transmit potentially adaptive non-genetically inherited phenotypes to F1 progeny”. While the effect on heat shock response seems larger, the associated decrease in *hsp* gene expression is also quite modest. These results are suggestive, but do not support a strong conclusion about intergenerational effects on phenotype.

*In our initial submission of this manuscript, we chose to display the Kaplan-Meier curve of the summation of our L1 survival data (5 independent biological replicates/experiments). While highly statistically significant (now **Supplementary Figure 5F** – $P = 9.83 \times 10^{-11}$), the amplitude of effect is subtle. However, in all 5 experiments we performed we found the same result, that *rnst-2* sired progeny displayed increased survival compared to WT controls which we now present as **Figure 5D**. The lack of a striking effect in the Kaplan–Meier curve, which averages the probability of survival across days, reflects variability in survival curves between experiments, for example, in some experiments L1s begin dying earlier than in others. Importantly, however, the extension of survival is consistently observed in progeny sired by *rnst-2* males.*

10) There are several problems with the anti-tDR injection experiment shown in Fig. 6. Fig. 6B shows a small and highly variable effect on *ife-2* expression, and Fig. 6C does not convincingly show rescue of the histone and *fbxb* genes compared to Fig. 5C. In addition, no data is shown to indicate how much anti-tDR was injected, or what effect this had on actual target tDR levels. The strong conclusions about the specific roles of

tDR-Gly-GCC and tDR-Glu-CTC in the Results and Discussion sections, including: “These findings demonstrate that tDR-Gly-GCC and/or tDR-Glu-CTC in sperm causally regulate early embryonic gene expression and non-genetically inherited phenotypes in offspring” and “we demonstrate that tDR-Gly-GCC and tDR-Glu-CTC are causal for the inheritance of these phenotypes” are not warranted based on the data shown. Overall, the study would be stronger with this set of experiments removed.

*While the regulation of *ife-2*, histone genes, and *fbxb* genes is less dramatic in our anti-tDR injections (**Figure 6**) compared to when we naturally mate *rst-2* males with WT females (**Figure 5**) the results are still significant statistically. We now provide Kolmogorov–Smirnov tests on our Cumulative Distribution Frequency plots (**Figure 6C inset**) which reveal highly significant shifts ($P > 0.0001$ for both), indicative of rescue, in the distribution of expression of histone and *fbxb* genes in 8-cell embryos sired by *rst-2* males and injected with anti-tDRs. However, we acknowledge that these results are not a complete rescue of the phenotypes, which we speculate result from the variability of the delivery of anti-tDR using this approach (see comment #8 above).*

*Overall, while we believe our results reveal causality for tDRs Gly-GCC and Glu-CTC in regulating post-fertilization gene expression and the transmission of non-genetically inherited phenotypes, we recognize that these experiments are not perfectly conclusive. To address this, we toned down our claims throughout the manuscript using words like ‘suggest’, ‘imply’, or ‘indicate’ rather than more definitive statements that were in the first version of our manuscript. Additionally, we have added a **Limitations of this Study** section to the discuss where we discuss this (and other) issues.*

11) The assertion that tDR function in sperm is “deeply conserved among metazoans” at the end of the introduction is a stretch based on only two species.

We now refrain from using “deeply conserved” throughout the manuscript.

12) In the first paragraph of the results, it would be useful to specify that previous studies were performed on whole adult hermaphrodite worms (vs. “adult hermaphrodite worms”), for readers who are accustomed to working with isolated adult mammalian tissues.

Thank you for pointing this out. We have made this change.

13) This pair of sentences in the first paragraph of the Results section is very confusing: “The *fog-2* mutation produces hermaphrodites that do not undergo spermatogenesis yet

produce normal oocytes. Further, fog-2 is not expressed or required for any aspect of male gametogenesis, thus producing normal sperm". Initially this appears contradictory, but on re-reading I think it means that hermaphrodites produce only oocytes, while males produce (only) normal sperm. If correct, this could be explained more clearly.

We have made this change. Thank you for improving the clarity of the manuscript!

14) In Fig. 3A, there are non-sperm GFP+ cells in the bottom image, which appears to contradict the claim in the text that GFP::RNST-2 is expressed specifically in sperm. I suspect that the source of the additional GFP is autofluorescence or a co-injection marker. If so, this should be specified in the legend. If it is from real transgene expression outside the reproductive tract, this should be more clearly explained in the text and the conclusion should be adjusted.

This signal is autofluorescence from gut granules, which often obscures GFP images in C. elegans. We have now pointed this out in the figure legend.

5) Please add page and line numbers to make comments easier to reference.

Done. We apologize for not including this in our first submission.

Reviewer #2 (Remarks to the Author):

It is known that in mouse models, the diet of males regulates production of specific tRNA-derived RNAs (tDRs) in sperm. While some mechanistic insights were obtained on the action of these tDRs in rodents, the details of their biogenesis are still missing. In this work, Galambos et al used worm model to get insights into similar phenomenon and determined the RNase T2 enzyme, *rnst-2*, as a regulator of *C. elegans* tDR accumulation. RNY1, yeast RNase T2, is shown to cleave tRNAs in the anticodon to produce tDRs. In worms, T2 does not cut the tRNA initially but regulates production of shorter tDRs, likely from tRNA halves.

The main question still remains- what is identity of the tRNase that makes first cut. Nonetheless, the data provided in this work is appealing.

My major concerns are that all of this work is based on RNA-seq. I would like to see a northern blotting and qRT-PCR to back up the data. The same is true for the experiments that are based on anti-tDR oligos, namely what is the efficiency of the depletion, what happens with mature tRNAs, are their levels affected?

*We thank the reviewer for raising this point. To address this, we have now included a Northern Blot for 5' Gly-GCC, 5' Glu-CTC, and U6 snRNA (loading control) on *rnst-2^{cat/cat}* and WT male RNA (**Figure 2F**). It was difficult to obtain sufficient material (5 µg RNA) for these experiments because *rnst-2* homozygous mutants exhibit developmental phenotypes that worsen over generations, which is the focus of a separate project in the lab. Consequently, within the timeframe of this revision we were only able to perform this analysis using the catalytic mutants. The results of our Northern Blotting analysis prominently support our small RNA-seq data from **Figure 2**.*

*Regarding the fate of tDRs in our anti-tDR experiments, it is important to note that anti-tDRs are not expected to decrease tDR abundance. Instead, they function by binding to tDRs and inhibiting their regulatory activity, blocking their ability to base-pair, much like miRNA inhibitors. This is now discussed both in our **Introduction** on Page 3, lines 96-104 and in the newly added **Limitation of the Study** section at the end of the discussion.*

The experiment with catalytically dead *rnst-2* is very appealing. Authors should use this mutant for "CLIP/RIP-like" experiments to demonstrate whether the enzyme directly pulls mature tRNAs or their cleavage variants, I believe that this would be very important to figure out.

*We thank the reviewer for acknowledging the strength of using an *rnst-2* catalytic mutant. We agree that biochemical analysis of *rnst-2* with tRNAs is the next step towards mechanistically understanding how tDR processing occurs in sperm. However, we tried for nearly a year to immunoprecipitate GFP tagged and endogenous RNST-2 with no success. RNST-2 is likely in an insoluble fraction of cellular lysates that precludes the experiments suggested by the reviewer. We acknowledge that this limits the conclusions we can make with our current study on the mechanism underlying tDR processing, which we now discuss in the **Limitations of the Study** section at the end of the discussion.*

Finally, the effect of tDR depletion on gene expression, especially on worm eIF4E are interesting. Would injection of synthetic tDRs and their mutants affect *ife-2*, histone and *fbxb* genes?

*This is another great point raised by the reviewer that it unfortunately limited experimentally. tRNAs are heavily chemically modified RNAs, and thus tDRs are as well. Several studies have demonstrated that RNA modifications are required for tDRs to function and thus required to recapitulate tDR associated phenotypes in gain-of-function experiments like injecting synthetic tDRs. Additionally, RNA modifications of tRNAs in *C. elegans* are very poorly understood, thus we would have no basis for synthetically designing tDRs with the appropriate modifications. We again acknowledge*

*that this is a limitation of this study, we now discuss this in the **Introduction** and **Limitations of the Study** section.*